# RNA editing of Filamin A pre-mRNA regulates vascular contraction and diastolic blood pressure

Mamta Jain[1],[*],[†], Tomer D Mann[2],[3],[†], Maja Stulić[1],[†], Shailaja P Rao[4], Andrijana Kirsch[4], Dieter Pullirsch[1], Xué Strobl[1], Claus Rath[5], Lukas Reissig[5], Kristin Moreth[6], Tanja Klein-Rodewald[6],[7], Raffi Bekeredjian[8], Valerie Gailus-Durner[6], Helmut Fuchs[6], Martin Hrabě de Angelis[6],[9],[10], Eleonore Pablik[11], Laura Cimatti[1], David Martin[1], Jelena Zinnanti[12], Wolfgang F Graier[4], Maria Sibilia[13], Saša Frank[4], Erez Y Levanon[2] (ID) & Michael F Jantsch[1],[**] (ID)

## Abstract

Epitranscriptomic events such as adenosine-to-inosine (A-to-I) RNA editing by ADAR can recode mRNAs to translate novel proteins. Editing of the mRNA that encodes actin crosslinking protein Filamin A (FLNA) mediates a Q-to-R transition in the interactive C-terminal region. While FLNA editing is conserved among vertebrates, its physiological function remains unclear. Here, we show that cardiovascular tissues in humans and mice show massive editing and that FLNA RNA is the most prominent substrate. Patient-derived RNA-Seq data demonstrate a significant drop in FLNA editing associated with cardiovascular diseases. Using mice with only impaired FLNA editing, we observed increased vascular contraction and diastolic hypertension accompanied by increased myosin light chain phosphorylation, arterial remodeling, and left ventricular wall thickening, which eventually causes cardiac remodeling and reduced systolic output. These results demonstrate a causal relationship between RNA editing and the development of cardiovascular disease indicating that a single epitranscriptomic RNA modification can maintain cardiovascular health.

Keywords adenosine deaminases acting on RNA (ADAR); cardiovascular disease; Filamin A (FLNA); hypertension; RNA editing
Subject Categories Development & Differentiation; RNA Biology; Vascular Biology & Angiogenesis

The EMBO Journal (2018) 37: e94813

## Introduction

Adenosine-to-inosine (A-to-I) RNA editing is the most prevalent epitranscriptomic change in mammalian RNAs (Nishikura, 2010). As most cellular machineries including translation interpret inosines as guanosines, A-to-I editing can recode mRNAs to produce the translation of novel proteins, not encoded in the genome (Nishikura, 2010; Pullirsch & Jantsch, 2010). A-to-I editing is catalyzed by adenosine deaminases acting on RNA (ADAR) that recognize double-stranded and structured RNAs (Nishikura, 2010). In mammals, ADAR1 and ADAR2 mediate all editing events. ADAR1 is expressed in all tissues and likely targets repeat-derived double-stranded (ds) RNAs. In contrast, ADAR2 shows its highest expression in the brain and can edit coding and non-coding regions of mRNAs (Riedmann et al, 2008; Nishikura, 2010). Most mammalian recoding edits known today affect mRNAs encoding ion channels and receptors within the central nervous system (Hoopengardner et al, 2003; Savva et al, 2012; Li & Church, 2013). Consequently, ADAR2-mediated recoding events were believed to mainly occur in nervous tissue. Impaired editing in humans is linked to neuronal disorders, type I interferonopathies, and cancer

1  Division of Cell Biology, Center for Anatomy and Cell Biology, Medical University of Vienna, Vienna, Austria
2  The Mina and Everard Goodman Faculty of Life Sciences, Bar Ilan University, Ramat-Gan, Israel
3  Tel Aviv Sourasky Medical Center, Tel Aviv, Israel
4  Center of Molecular Medicine, Institute of Molecular Biology and Biochemistry, Medical University of Graz, Graz, Austria
5  Division of Anatomy, Center for Anatomy and Cell Biology, Medical University of Vienna, Vienna, Austria
6  German Mouse Clinic, Institute of Experimental Genetics, Helmholtz Zentrum München, Neuherberg, Germany
7  Institute of Pathology, Helmholtz Zentrum München, Neuherberg, Germany
8  Department of Cardiology, University of Heidelberg, Heidelberg, Germany
9  Department of Experimental Genetics, Center of Life and Food Sciences Weihenstephan, Technische Universität München, Freising-Weihenstephan, Germany
10 German Center for Diabetes Research (DZD), Neuherberg, Germany
11 Section for Medical Statistics, CeMSIIS, Medical University of Vienna, Vienna, Austria
12 Vienna Biocenter Core Facilities GmbH, Vienna, Austria
13 Department of Medicine I, Comprehensive Cancer Center, Institute for Cancer Research, Medical University of Vienna, Vienna, Austria
    *Corresponding author. Tel: +43 1 40160 37752; E-mail: mamta.jain@meduniwien.ac.at
    **Corresponding author. Tel: +43 1 40160 37510; Fax: +43 1 40160 37542; E-mail: michael.jantsch@meduniwien.ac.at
    †These authors contributed equally to this work

(Paz *et al*, 2007; Rice *et al*, 2012; Chen *et al*, 2013; Paz-Yaacov *et al*, 2015).

One conserved mammalian editing substrate encodes the actin crosslinking protein Filamin A (FLNA; Stossel *et al*, 2001; Levanon *et al*, 2005). FLNA is composed of 24 Ig-like domains organized in two rod-regions separated by a hinge (Fig EV1). FLNA homo- and heterodimerizes with the paralogous protein FLNB via its 24th C-terminal Ig-repeat, while the N-terminal region mediates actin binding (Stossel *et al*, 2001; Robertson, 2005). Loss of FLNA in mice causes vascular abnormalities and reduced vascular tension (Feng *et al*, 2006; Hart *et al*, 2006; Retailleau *et al*, 2016). Exon 42 editing induces a Q-to-R amino acid exchange in Ig-repeat 22 in a region that can interact with over 90 proteins (Stossel *et al*, 2001; Levanon *et al*, 2005; Zhou *et al*, 2010; Nakamura *et al*, 2011; Fig EV1). In mice, FLNA editing primarily occurs in the vasculature and the digestive tract, which makes FLNA the first prominent recoding event outside the nervous system (Levanon *et al*, 2005; Stulic & Jantsch, 2013).

Here, using large-scale publically available control and patient transcriptome data sets, we show that FLNA editing mediated by ADAR2 in human cardiovascular tissues exceeds the total ADAR2 editing activity previously identified in nervous tissue making it the prime editing target. Importantly, samples derived from cardiovascular patients show a dramatic reduction in FLNA editing in cardiovascular tissues. To explore the function of FLNA editing in the cardiovascular system, we generated transgenic mice impaired in FLNA editing. These mice show increased vascular contraction, elevated blood pressure, arterial remodeling, and left ventricular wall thickening, which eventually leads to left ventricular hypertrophy and cardiac remodeling. Thus, we establish the biomedical impact of a single RNA editing event and reveal a putative biomarker or therapeutic handle.

## Results

### FLNA editing is highest in cardiovascular tissue and significantly reduced in patients with cardiac disease

mRNA recoding by ADAR2 is typically a brain-specific phenomenon that can diversify receptor function (Holmgren & Rosenthal, 2015). However, the availability of large-scale transcriptome data permits detailed analyses to revisit this long-held assumption. The GTEx database offers high-quality transcriptome data from dozens of tissues from hundreds of donors (GTEx Consortium, 2013), which enables a comprehensive survey for the expression and activity of editing enzymes. Unexpectedly, we found the highest expression levels of ADAR2 in the tibial artery, aorta, coronary arteries, and other vascular tissues, far exceeding the previously reported prominent ADAR2 expression in the nervous system (Fig 1A and Appendix Fig S1; Melcher *et al*, 1996). We found ADAR1 and ADAR3 expressions match the previously reported ubiquitous and prevalent neuronal expressions, respectively (Kim *et al*, 1994; Chen *et al*, 2000; Appendix Fig S1).

We then used the GTEx RNA-Seq data to comprise a list of 252 putative editing sites located in coding sequences to identify the main editing targets in the cardiovascular system (Table EV1). A cluster analysis of editing substrates and levels within the

cardiovascular system demonstrates that the appendage and ventricle cluster together, as well as tibial artery, dorsal aorta, and coronaries (Appendix Fig S3). Among these candidates, we found Filamin A was highly edited, up to 98% in the aorta and coronary and tibial arteries (Fig 1C and Appendix Fig S2), in which FLNA has an extremely high expression level (Fig 1B and Appendix Fig S2). When total A-to-I editing levels were further compared for the 38 most highly edited sites between vascular and nervous tissue (cerebellum), much higher editing levels were found in vascular than in neuronal tissue (Appendix Fig S4A). In fact, editing of Gria2, the previously considered most abundant ADAR2 target, only ranks at position 9, almost two orders of magnitude below FLNA (Appendix Fig S4B and Appendix Table S1A and B). This shows that editing in coding regions is most abundant in the vascular system, massively affecting previously less considered substrates such as FLNA or IGFBP7. So the FLNA transcript seems a prime substrate of ADAR2 across all tissues, exceeding the editing reactions at all previously known brain-related sites. Further, FLNA editing and ADAR2 expression show a good correlation in the vasculature (Fig 1, and Appendix Fig S4 and Appendix Table S1, and Table EV1).

To assess the putative impact of these abundant editing events in vascular tissues on cardiovascular health, we analyzed the five cardiovascular tissues available in GTEx (aorta, tibial artery, coronaries, left ventricle, and left atrial appendage yielding 1,111 samples derived from 478 donors) for sites of robust A-to-G editing. We divided this cohort into 268 donors with cardiovascular conditions and 210 donors without cardiovascular conditions according to the provided health records. We also scanned NCBI's SRA databank for sequencing data from heart failure patients and recognized a suitable cohort (Schafer *et al*, 2017). We then compared editing rates between these cardiomyopathy patients and the healthy GTEx cohort and selected editing sites that demonstrated remarkable (> 1.9 fold) differences in editing rates between healthy subjects and patients with dilated cardiomyopathy (DCM; Appendix Table S2). Doing so, the most significant change in editing was found in the RNA encoding SON. Here, editing leads to a synonymous codon exchange, which will not affect the encoded protein.

The most significant change in editing leading to a non-synonymous codon exchange occurred at the conserved editing site in Filamin A, which gives rise to a Q2341R amino acid exchange (Fig 1D). Average editing levels in FLNA mRNA dropped by half from 23% in 69 samples to 12% in 112 heart samples of patients suffering from dilated cardiomyopathy ($P = 2.237471e\text{-}06$) (Schafer *et al*, 2017) in the SRA cohort when compared to the unaffected GTEx samples (Fig 1D). We found that FLNA was also a significant discriminator between healthy individuals and those suffering from a cardiovascular disease within the GTEx cohort itself (23 vs. 17%, respectively, $P = 0.003$, Table EV2).

Filamin A editing levels in the vascular tissue are even higher than in the ventricle. Thus, we compared editing rates from tibial arteries and aortae from fresh cadavers with visible cardiac hypertrophy or aneurisms with those from cadavers with no record or signs of cardiac pathologies (LV hypertrophy, aneurism). Interestingly, a striking difference was observed in tibial arteries where average editing rates dropped from

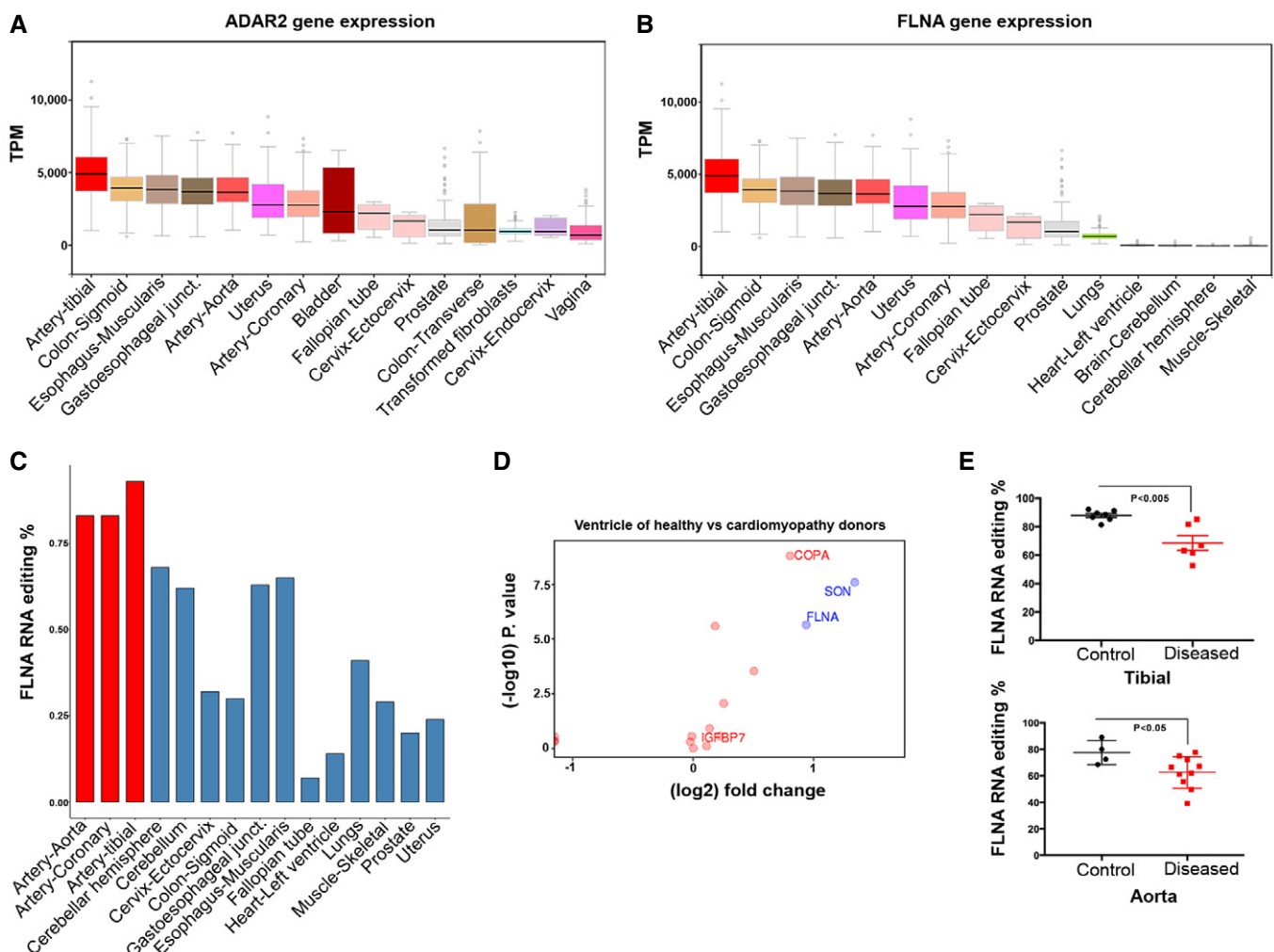

**Figure 1. FLNA editing in cardiac patients from GTEx donors.**

A   ADAR2 is most strongly expressed in vascular tissues. Graph showing ADAR2 gene expression data derived from GTEx for few representative human tissues. Note the highest expression of ADAR2 in the tibial artery followed by colon and esophagus. Boxes represent the 25[th] and the 75[th] percentile with median represented by the black line in the box. The whiskers depict the minimum and the maximum value.

B   FLNA gene expression in few representative human tissues from > 500 different donors using GTEx data. Tibial artery, colon, and esophagus show highest gene expression similar to ADAR2 expression. Boxes represent the 25[th] and the 75[th] percentile with median represented by the black line in the box. The whiskers depict the minimum and the maximum value.

C   Bar graph shows FLNA RNA editing (%) among few representative human tissues. Note very high editing levels in the arterial system (red bars).

D   Scatter plot shows the log fold change in editing levels of several candidates between ventricles of healthy donors and heart samples of cardiomyopathy patients. FLNA marked in the graph is one significant discriminator. Y-axis is the −log 10 of the P-value for the difference between healthy and sick. The colors (red vs. green) reflect the threshold cutoffs randomly assigned to distinguish the sites, which demonstrate large, highly significant differences between the two groups. Fold change cutoff of 0.9 was used.

E   Scatter plot shows the FLNA RNA editing in control and diseased human tibial arteries and aortae. At least four control and seven diseased human donors were used in each case. P-value < 0.05 measured by t-test was considered significant.

Source data are available online for this figure.

87.8 ± 3.7% in control samples to 68.5 ± 12.5% in donors showing dilated cardiomyopathy due to cardiovascular disease while editing rates in aortae dropped from 77.5 ± 9.1 to 62.6 ± 11.8% (Fig 1E). However, it should also be noted that editing levels stayed normal in some samples showing cardiac pathologies indicating that not every cardiac pathology is accompanied by a drop in FLNA editing. Next, we tested whether the drop in editing levels could be correlated with ADAR2 expression levels. As shown in the correlation regression plots in Appendix Fig S4C, no significant correlation was found between ADAR2 expression and FLNA editing in tibial artery and aorta from healthy and diseased cadavers. This is in contrast to the GTEx data where a good correlation between the two parameters was seen when different tissues were compared (Fig 1A–C). Taken together, these data demonstrate a correlation between a drop in FLNA editing and cardiovascular pathologies.

## Creation of mice deficient in FLNA editing

We then sought to determine whether changes in Filamin A editing have a causal effect on the development of cardiovascular pathologies. We generated a mouse with an exclusive deficit in FLNA editing. We disrupted the double-stranded structure required for editing by removing the editing complementary site (ECS) in intron 42 from the X-linked *Filamin A* gene via homologous recombination in ES cells (Fig 2A). The resulting hemizygous males and homozygous females were devoid of FLNA mRNA editing (Fig 2B). These mice designated as FLNA$^{\Delta ECS}$ did show normal FLNA expression levels as judged by qPCR (Fig 2C), Western blot (Fig 2D), and RNA-Seq (Appendix Fig S5A). As FLNA and FLNB can heterodimerize and have similar functions, we also examined the expression and editing of FLNB. However, neither FLNB expression nor FLNB editing was significantly affected in the absence of FLNA editing (Appendix Fig S5B). We observed no apparent abnormalities and normal life expectancy and fertility in both male and female FLNA$^{\Delta ECS}$ mice. Due to the X-linked nature of the FLNA gene, we primarily examined male littermates, which only carry a single allele for further analysis.

## Absence of FLNA editing increases smooth muscle contraction

In mice and humans, FLNA editing is highest in vascular tissues and other organs rich in smooth musculature (Stulic & Jantsch, 2013; Fig 2). Separation of mouse dorsal aortae verified that the smooth muscle layer (tunica media) displays significant FLNA editing levels (> 90%) compared to the 60% editing levels in the whole aorta (Fig 2E). Since FLNA crosslinks actin and interacts with regulators of smooth muscle cell contraction, such as RhoA and ROCK (Nakamura *et al*, 2011), we tested whether FLNA editing affects smooth muscle contraction. We tested rings of dorsal aortae in myograph chambers. Indeed, FLNA$^{\Delta ECS}$ aortic rings showed a significant increase in smooth muscle contraction upon treatment with the thromboxane A2 receptor agonist U46619 compared to wild-type (wt) mice (Fig 3A). Even though E$_{max}$ increased, the EC50 remained unaltered (Fig 3A). Both maximum contraction and contractile force increased by 30% in aortae of FLNA$^{\Delta ECS}$ mice (Fig 3B). After U46619 administration, primary vascular smooth muscle cells (vSMCs) from FLNA$^{\Delta ECS}$ mice also hypercontracted compared to wt vSMCs with a comparable dose–response (Fig 3C and D). Together, these results indicate that FLNA editing regulates the contraction of smooth muscle cells.

Thromboxane A2 receptor-mediated smooth muscle cell contraction can signal through ROCK kinase, activated by G12/13, or PLC/PKC, activated by Gq/11 (Wettschureck & Offermanns, 2005). We used inhibitors for ROCK (Y27632), PKC (GF109203X), PLC (U-73122), and Ca$^{2+}$ signaling (BAPTA) to identify pathway-specific differences (Fig EV2). All inhibitors had a strong general effect on aortic ring contraction and led to a reduced relative contraction upon U46619 stimulation. Inhibition of ROCK, PKC, and PLC alleviated genotype-specific contractile differences. This suggests that editing-dependent contractile differences only manifest upon maximum contractibility but diminish when overall contraction is impaired (Fig EV2). ROCK inhibition by Y27632 on pre-contracted aortic rings enhanced relaxation, consistent with increased ROCK activity in FLNA$^{\Delta ECS}$ mice (Fig EV2B). Filamin A can mediate

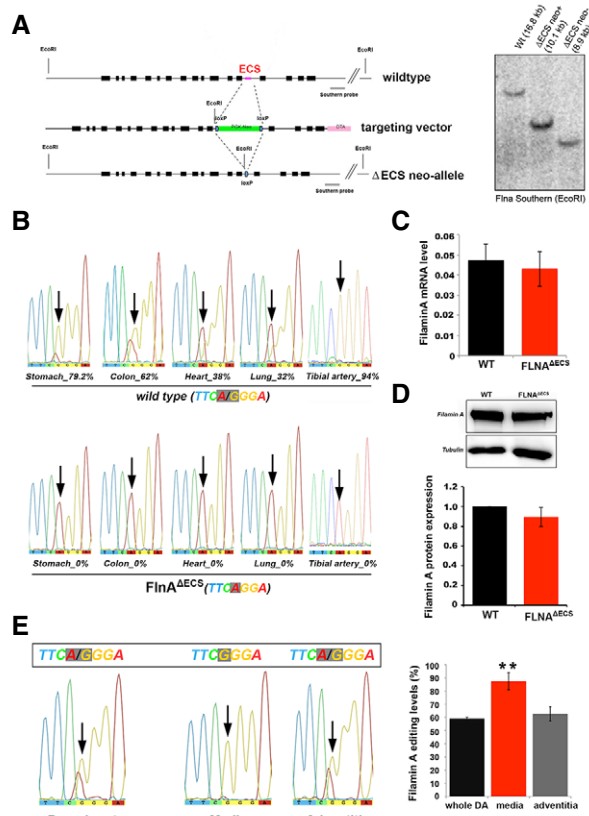

**Figure 2.  Generating mice deficient in Filamin A editing.**

A  Scheme showing the FLNA wild-type (wt) allele, the targeting vector, and the FLNA$^{\Delta ECS}$ neo-allele. The loxP flanked PGK-neo cassette in the targeting vector replaced the editing complementary sequence (ECS) using homologous recombination, which was then deleted using Cre recombinase. Right, Southern blotting analysis screened for positive clones shown. The positions of loxP sites, restriction enzyme (EcoRI), and Southern blotting probe are also indicated.

B  Sequencing electropherograms show the average FLNA editing levels in wt and FLNA$^{\Delta ECS}$ tissues (stomach, colon, heart, lung, and tibial artery). Editing levels were checked in tissues from three independent mice, and the value below the chromatogram depicts the average value of three replicates.

C  FLNA mRNA expression levels measured by qPCR in wt and FLNA$^{\Delta ECS}$ colon tissue showed no difference.

D  FLNA protein levels measured by Western blotting are identical in wt and FLNA$^{\Delta ECS}$ stomach tissue. *Y*-axis represents FLNA expression normalized to tubulin levels.

E  FLNA editing levels (%) in wt whole dorsal aorta, tunica media, and tunica adventitia. Tunica media consisting of smooth muscle cells show the highest FLNA RNA editing.

Data information: Arrows indicate the FLNA editing site. For (C–E), data are shown as mean ± SD from three independent experiments. **$P$ < 0.05. Source data are available online for this figure.

tension-specific regulation of Ca$^{2+}$ channels (Retailleau *et al*, 2016). Indeed, chelation of intracellular calcium using BAPTA-AM abolished genotype-specific differences, indicating that FLNA editing can affect mechanotransduction and hence intracellular Ca$^{2+}$ levels (Fig EV2E). Together, these experiments demonstrate that lack of FLNA RNA editing enhances stimulation of thromboxane A2 receptor-mediated cellular contraction.

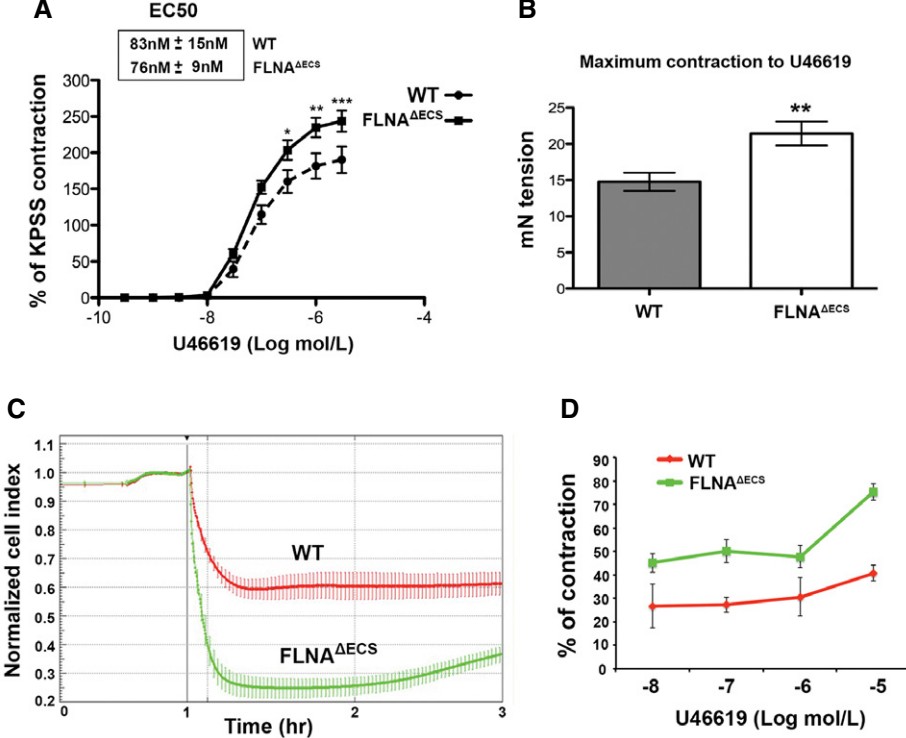

**Figure 3.  FLNA^ΔECS aortae and primary vSMCs show hypercontraction in response to U46619.**

A, B    Treatment of aortic rings with thromboxane A2 receptor agonist U46619 leads to (A) hypercontraction and (B) increased contraction force in FLNA^ΔECS aortae in myograph chambers. Emax was higher in FLNA^ΔECS aortae without any difference in EC50 as compared to wt aortae. For each condition, 10–12 aortic rings from at least four wild-type (wt) and four FLNA^ΔECS mice were used. Data are shown as mean ± SEM. *P < 0.05, **P < 0.01, ***P < 0.001 (Student's t-test)
C       Graph shows the normalized cell contraction measured by xCELLigence Real-Time Cell Analyzer in wt (red) and FLNA^ΔECS (green) primary vascular smooth muscle cells (vSMCs) indicating hypercontraction in FLNA^ΔECS cells in response to 10 μM U46619. Data are shown as mean ± SD from three independent experiments.
D       Quantification of cell index measurements plotted as percentage of contraction following different concentrations of U46619. Data are shown as mean ± SD from three independent experiments. P-value < 0.05 (Student's t-test) was considered significant. The difference between wt and FLNA^ΔECS was significant at every concentration of U46619.

Source data are available online for this figure.

## Lack of FLNA editing induces elevated myosin phosphorylation

Smooth muscle contraction ultimately depends on myosin light chain (MLC) phosphorylation. Indeed, MLC phosphorylation (pMLC) is increased in FLNA^ΔECS vSMCs (Fig 4A). RhoA GTPase activates ROCK kinase to control MLC phosphorylation (Totsukawa *et al*, 2000). Consistently, we found increased levels of activated RhoA-GTP in FLNA^ΔECS vSMCs compared to wt cells (Fig 4B). MLC dephosphorylation controls smooth muscle relaxation. MLC dephosphorylation is mediated by myosin light chain phosphatase 1 (MYPT1), which itself is inhibited by ROCK-mediated phosphorylation. So increased MYPT1 phosphorylation is another hallmark of increased ROCK activity and is an indicator of SMC contraction. Consistently, FLNA^ΔECS vSMCs show increased MYPT1 phosphorylation, which is further enhanced after addition of the thromboxane A2 receptor agonist U46619 (Fig 4C).

MYPT1 is inhibited by CPI-17, which in turn is inhibited by phosphorylation by both ROCK and PKC (Shibata *et al*, 2015). CPI-17 phosphorylation is increased in vSMCs derived from FLNA^ΔECS mice (Fig 4D). Inhibition of either ROCK or PKC abolished

genotype-specific differences in CPI-17 phosphorylation. These results demonstrate that FLNA editing modulates both branches of thromboxane A2 signaling (Fig 4D). Together, these data suggest that all critical factors for smooth muscle contraction and MLC phosphorylation are upregulated in FLNA^ΔECS vSMCs and that both ROCK and PKC signaling contribute to this phenomenon.

Next, we tested for changes in expression or localization of known FLNA interactors that can modulate RhoA activity. Of these, p190^RhoGAP showed a distinct cellular localization in FLNA^ΔECS cells (Mammoto *et al*, 2007; Fig EV3). p190^RhoGAP is a *bona fide* RhoA GTPase activating protein that can inhibit ROCK activation by promoting GTP hydrolysis of RhoA (Mori *et al*, 2009; Puetz *et al*, 2009; Bhattacharya *et al*, 2014). Conversely, angiotensin and endothelin can increase RhoA-GTP levels by decreasing p190^RhoGAP activity (Bouchal *et al*, 2009; Bregeon *et al*, 2009). To promote RhoA GTPase activity, p190^RhoGAP must be localized in close proximity to RhoA at the plasma membrane. FLNA can regulate the membrane localization of p190^RhoGAP (Mammoto *et al*, 2007; Oinuma *et al*, 2012). So we determined the localization of p190^RhoGAP in wild-type and FLNA^ΔECS vSMCs. We observed

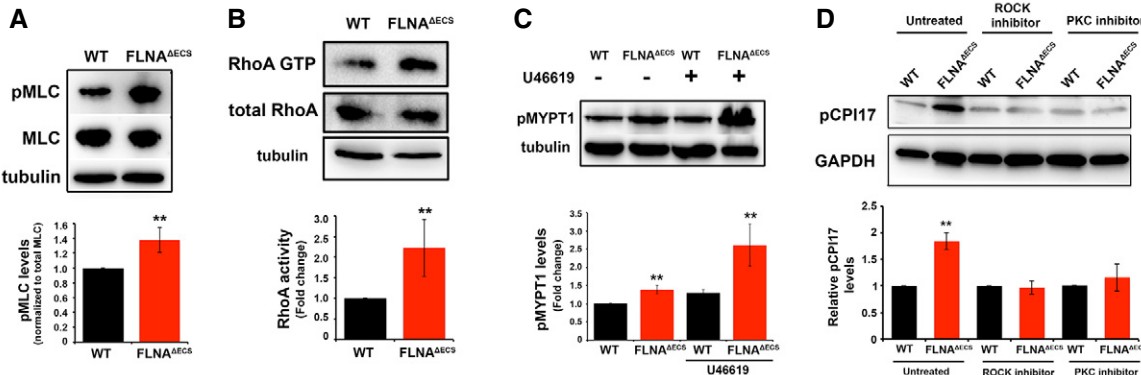

**Figure 4. FLNA$^{\Delta ECS}$ vSMCs have increased Rho and ROCK activity and myosin phosphorylation.**

A   Myosin light chain phosphorylation detected by a phospho-specific antibody (pMLC) and normalized to total myosin light chain (MLC). Tubulin served as a loading control. Quantification shows a significant increase in pMLC in FLNA$^{\Delta ECS}$ vSMCs.

B   Rhotekin pull-down activation assay was performed on wt and FLNA$^{\Delta ECS}$ vSMC lysates, and activated GTP-bound RhoA was compared between the two genotypes. FLNA$^{\Delta ECS}$ vSMCs show more GTP-loaded RhoA than wt vSMCs. Tubulin was used as a loading control.

C   Thromboxane A2 stimulation with U46619 increased phosphorylation of myosin light chain phosphatase (pMYPT1) in FLNA$^{\Delta ECS}$ vSMCs as shown by Western blotting on vSMC lysates with phospho-specific MYPT1 antibody.

D   CPI17 phosphorylation levels were measured in wt and FLNA$^{\Delta ECS}$ vSMCs in untreated cells and in the presence of ROCK and PKC inhibitor using pCPI17 antibody.

Data information: Data are shown as mean ± SD from at least three independent experiments. **$P < 0.05$ (Student's $t$-test).
Source data are available online for this figure.

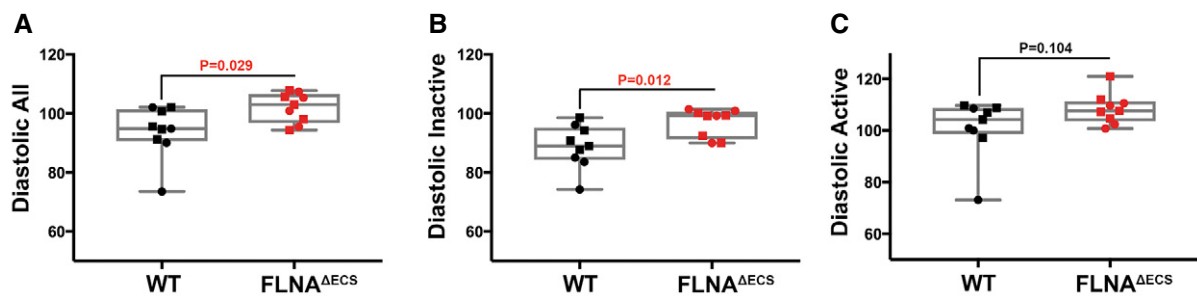

**Figure 5. FLNA$^{\Delta ECS}$ mice are hypertensive during resting periods.**

A–C   Telemetry measurements were done on mice continuously for 72 h, and reading was recorded every minute to analyze blood pressure in wt and FLNA$^{\Delta ECS}$ mice. Scatter plots show diastolic blood pressure over 72 h (A), diastolic blood pressure during inactive phase (B), and diastolic pressure during active phase (C). Note: Significant increase in diastolic blood pressure in FLNA$^{\Delta ECS}$ mice during the inactive phase. Nine mice (five males marked as square + four females marked as round) were analyzed for each genotype. Data are shown as mean ± SEM. $P$-value $< 0.05$ (Student's $t$-test) was considered significant. Boxes represent the 25[th] and the 75[th] percentile with median represented by the black line in the box. The whiskers depict the minimum and the maximum value.

Source data are available online for this figure.

distinctly more membrane localization in wild-type vSMCs than in FLNA$^{\Delta ECS}$ cells (Fig EV3). Addition of U46619 to vSMCs redistributed most p190$^{RhoGAP}$ to the cytoplasm (Fig EV3). This result is consistent with the notion that cortical localization of p190$^{RhoGAP}$ inhibits Rho-GTP levels and that FLNA$^{\Delta ECS}$ cells show strongly reduced cortical localization of p190$^{RhoGAP}$.

To determine changes in the FLNA interactome induced by the editing-induced Q-to-R amino acid exchange, we immunoprecipitated FLNA from wt and FLNA$^{\Delta ECS}$ vSMCs followed by quantitative mass spec analysis. About 300 interactors could be consistently detected in three replicates (Fig EV4 and Table EV3). When normalized to FLNA mass spec counts or to the mean of IP MS counts, about 20 proteins were found specifically enriched in the IP of

FLNA$^{\Delta ECS}$ while about 30 proteins were found enriched in IPs of wt FLNA. Consistent with the stronger contractility of vSMCs expressing FLNA$^{\Delta ECS}$, actin and actin-associated proteins were enriched on the FLNA$^{\Delta ECS}$ IP (Fig EV4, yellow in Table EV3). More nuclear proteins were found associated with editable FLNA, suggesting a nuclear role of FLNA or of its C-terminal fragment as previously reported (Zheng *et al*, 2014).

**FLNA$^{\Delta ECS}$ mice show diastolic hypertension**

Vascular smooth muscle contraction affects blood pressure (Michael *et al*, 2008). Since we wanted to study the physiological consequences of vascular hypercontraction in FLNA$^{\Delta ECS}$ mice, we

performed long-term, blinded, telemetric *in vivo* blood pressure measurements in conscious wt and FLNA$^{\Delta ECS}$ male mice, while simultaneously recording heart rate and activity profiles (Huetteman & Bogie, 2009; Fig EV5D and E). We found the average diastolic blood pressure, which is controlled by vessel contraction, significantly increased over a 72-h time period in FLNA$^{\Delta ECS}$ mice compared to wt mice (Fig 5A). In contrast, systolic pressure, mainly governed by cardiac output, was comparable in both genotypes (Fig EV5A). We observed the most significant increase in diastolic blood pressure in mutant mice during inactive periods when mice were at rest (Fig 5B), whereas the diastolic blood pressure during the active phase remained unchanged (Fig 5C). We observed no major differences in systolic blood pressure between tested genotypes during active or inactive phase (Fig EV5B and C). These results show that FLNA editing is essential to lower diastolic blood pressure during phases of relaxation.

## Arterial remodeling and cardiovascular abnormalities in FLNA$^{\Delta ECS}$ mice

Continuous high blood pressure strains aortic vessels and can induce changes in vessel diameter, wall thickening, and increased collagen deposition, eventually causing vessel stiffness (Erdogan *et al*, 2007). A long-term increase in blood pressure and pressure overload in blood vessels can lead to cardiac remodeling (Kenchaiah & Pfeffer, 2004; Renna *et al*, 2013). Histological analyses of aortae isolated from 5- to 6-month-old mice showed a significant and consistent increase in collagen deposition in the adventitia (Fig 6A and B) in FLNA$^{\Delta ECS}$ compared to wt mice.

Histological analyses of hearts obtained from FLNA$^{\Delta ECS}$ and wt mice by Masson's trichrome staining revealed increased perivascular fibrosis in coronary vessels of FLNA$^{\Delta ECS}$ mice (Fig 6C and D). Perivascular fibrosis has indeed been correlated with increased ROCK activity (Rikitake *et al*, 2005; Shimizu & Liao, 2016). Further, H&E analysis on heart longitudinal sections showed a significant increase in the left ventricular wall of mutant mice at 5–6 months of age (Fig 6E and F), whereas the interventricular septum thickness did not change significantly in mutant mice compared to wild type (Fig 6E and F). WGA staining of heart cross-sections showed a significant increase in cardiomyocyte diameter in FLNA$^{\Delta ECS}$ hearts as compared to controls (Fig 6G and H). These results indicate that FLNA$^{\Delta ECS}$ hearts compensate for increased blood pressure by increasing cardiomyocyte diameter and collagen deposition around coronaries and aortae. The microvasculature is a major determinant of blood pressure. We therefore tested whether changes could be observed on the microvasculature of the kidney. We therefore stained arterioles in the renal cortex and measured the wall thickness of these vessels. Interestingly, no significant remodeling was observed in the kidney microvasculature when compared between FLNA$^{\Delta ECS}$ and wt mice (Appendix Fig S8). However, considering that editing levels in the overall kidney is only at 20% (Stulic & Jantsch, 2013) it is possible that editing levels in the microvasculature are generally low, therefore showing no effect between wild-type and FLNA$^{\Delta ECS}$ vessels.

To test whether the observed cardiac phenotypes were a secondary effect to hypertension or would also appear autonomously, we tested younger 21-day-old mice for the appearance of either perivascular fibrosis or enlarged cardiomyocytes. However, in

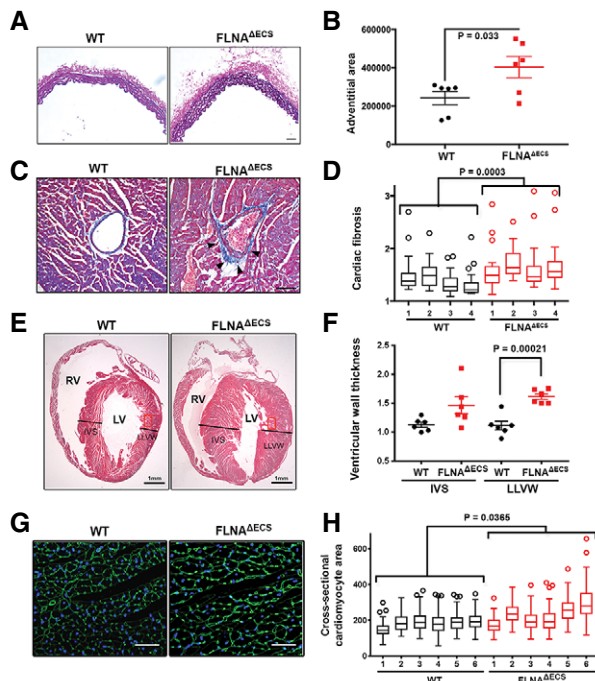

**Figure 6.  FLNA$^{\Delta ECS}$ mice show arterial and cardiac remodeling.**

A  Dorsal aortae were fixed, cross-sectioned, and then stained with Elastica van Gieson kit to visualize collagen around the blood vessel. Six independent mice (5–6 months old) of each genotype were used for the analysis. Scale bar: 50 μm.

B  Plot shows individual data points (black—wt, red—mutant) of adventitial area in both the genotypes. Measurements show increased adventitial area in FLNA$^{\Delta ECS}$ aortae. *P*-value < 0.05 (Student's *t*-test) was considered significant.

C  Representative heart sections of a 5- to 6-month-old mice stained with Masson's trichrome kit show increased collagen (blue) around coronary vessels (marked by arrowheads). Scale bar: 50 μm.

D  Quantification shows a significant increase in perivascular fibrosis in FLNA$^{\Delta ECS}$ hearts. Four sections from each heart were measured. Four mice analyzed for each genotype. *P*-value < 0.05 (mixed model approach) was considered significant.

E  Representative longitudinal heart sections stained with H&E (RV, right ventricle; LV, left ventricle). Black lines represent thickness of interventricular septum (IVS) and lateral left ventricular wall (LLVW). Scale bar: 1 mm.

F  Scatter graph shows increased ventricular wall thickness in FLNA$^{\Delta ECS}$ mice hearts. Six male mice (6 months old) analyzed for each genotype. *P*-value < 0.05 (Student's *t*-test) was considered significant.

G  Heart cross-sections were stained with FITC-WGA (wheat germ agglutinin) to visualize cell membranes of cardiomyocytes. Scale bar: 20 μm.

H  Respective cell areas measured by ImageJ software and represented by box plots for each animal (six per genotype). Boxes represent the 25th and the 75th percentile with median represented by the black line in the box. The whiskers depict the minimum and the maximum value. *P*-value < 0.05 (mixed model approach) was considered significant.

Source data are available online for this figure.

contrast to 5-month-old mice, no signs of abnormal cardiac organization were detected (Appendix Fig S7).

Using MRI, we also measured the systolic output of hearts in 24-month-old mice. We took measurements at the sinotubular junction at the beginning of the ascending aorta (Fig 7). Interestingly, FLNA$^{\Delta ECS}$ mice showed a significantly reduced cardiac output, a

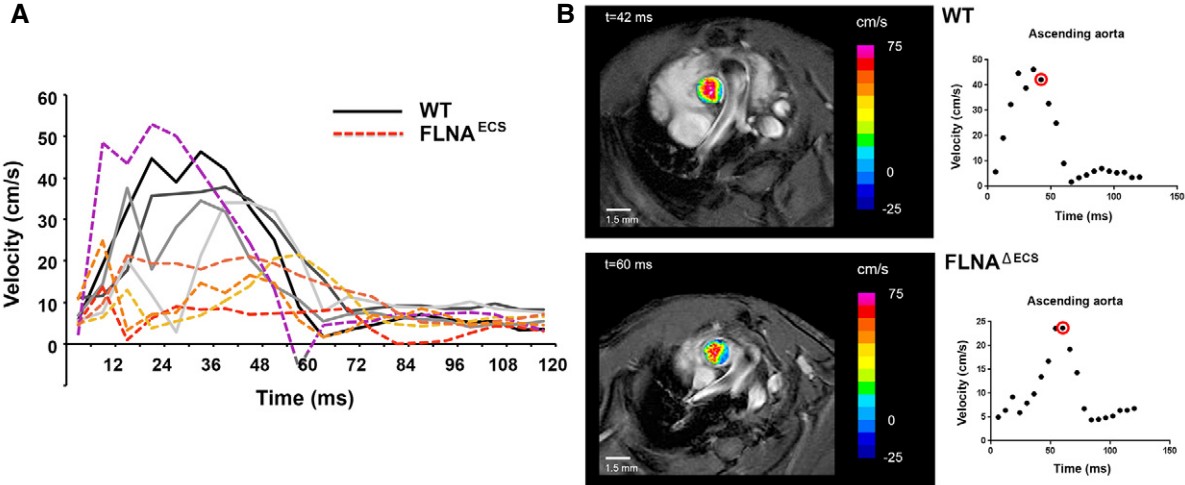

**Figure 7. Blood velocity profiling in FLNA$^{\Delta ECS}$ and wt mice.**

A  Graph shows the individual blood velocity profile determined by MRI over time in 4 wt mice (shades of gray) and 5 FLNA$^{\Delta ECS}$ mice (shades of red). The mouse shown in dotted purple line is clearly an outlier within the FLNA$^{\Delta ECS}$ cohort. Aortic velocity waveform represents velocity during different stages of cardiac cycle. A significant reduction in velocity was observed at 40 ms in the systolic cycle.

B  Representative wt (top) and FLNA$^{\Delta ECS}$ (below) mice showing the blood flow just above the aortic valve. Note: Reduced blood flow in FLNA$^{\Delta ECS}$ mouse as depicted by pseudocoloring. The red circle represents the time window at which the images are shown.

hallmark of heart failure depicted by blood velocity profile (Fig 7A) and representative MRI images (Fig 7B). Taken together, these data demonstrate that the absence of FLNA editing in vascular smooth muscle cells is a major risk factor for elevated diastolic blood pressure and subsequent heart pathology.

## Discussion

Internal and external cues by chemical modifications regulate transcriptome responses. Adenosine-to-inosine deamination is the most prevalent epitranscriptomic change yet known. This modification can recode mRNAs to translate novel protein variants, which are primarily known in the brains of higher metazoa (Dillman *et al*, 2013). To date, ADAR2 is thought to act mainly in the brain, and its clinical significance has thus far been linked to nervous system-related functions such as AMPA receptor editing, which, if disturbed, leads to intractable seizures and death. The analysis performed here, however, demonstrates that ADAR 2 activity in vascular tissue far exceeds editing activity in the brain. ADAR2 expression and activity in arteries are 10 times higher than those in the cerebellar hemisphere, probably the location of highest editing activity in the nervous system. Also, when examining total editing by site, the amount of editing taking place in IGFBP7 and FLNA in the vascular system is more than 100-fold larger than that in the brain expressed GluR2, which so far was considered the key ADAR2-mediated RNA editing site. Our results therefore suggest that ADAR2 does not primarily act in the nervous system, but rather the cardiovascular system, where editing in sites such as FLNA plays a key role in regulating vascular constriction, thereby protecting against cardiac remodeling and resulting heart disease. Surveying the conserved sites of RNA editing in the brains of mammals, it is apparent that the majority of ADAR2 editing takes place in arteries, 30% of which

affects FLNA. Other sites, which were recognized as key targets of ADAR2, are IGFBP7, SON, and COPA. Out of which, IGFBP7 seems a very promising site for future research, both for its extremely high editing activity and for its past implication in vascular pathologies such as aneurysm formation and pulmonary stenosis.

Editing of FLNA exceeds editing levels of 90% in mice and humans (Stulic & Jantsch, 2013). So the editing-induced Q2341R variant is the most abundant Filamin version in those tissues. By mining GTEx and SRA data, we detected a strong coincidence between a drop in FLNA editing and the prevalence of cardiovascular diseases. Using a transgenic mouse model, we demonstrate that the absence of FLNA editing causes hypercontraction of smooth muscle cells, inhibition of normal aortic relaxation, and increased diastolic blood pressure (Fig 8). This, in turn, also promotes remodeled aorta, enlarged cardiomyocytes and possible cardiomyopathy, a major heart pathology, and a leading risk factor for cardiac death. Thus, our results demonstrate that decreased FLNA editing not only correlates with cardiovascular diseases, but also can also contribute to cardiovascular diseases.

Under normal conditions, smooth muscle cells promptly adapt and change contraction with blood flow and shearing stress (Gabella, 1976; Yamin & Morgan, 2012). A complete deletion of FLNA from smooth muscle cells can affect blood pressure control presumably by controlling $Ca^{2+}$ influx via the stretch-activated cationic channel Piezo1, since angiotensin-mediated contraction seems unaffected (Retailleau *et al*, 2016). Interestingly, PKC, PLC, and ROCK inhibitors alleviate the increased contraction observed in editing-deficient FLNA$^{\Delta ECS}$ aortae (Fig EV2). BAPTA treatment abolished genotype-specific differences indicating that the observed differences in contraction likely depend on intracellular $Ca^{2+}$ signaling.

Filamin A interacts with many signaling proteins, including RhoGEFs, GAPs, Rho GTPase, or ROCK kinase, so FLNA can modulate smooth muscle contraction through several pathways (Fig EV1;

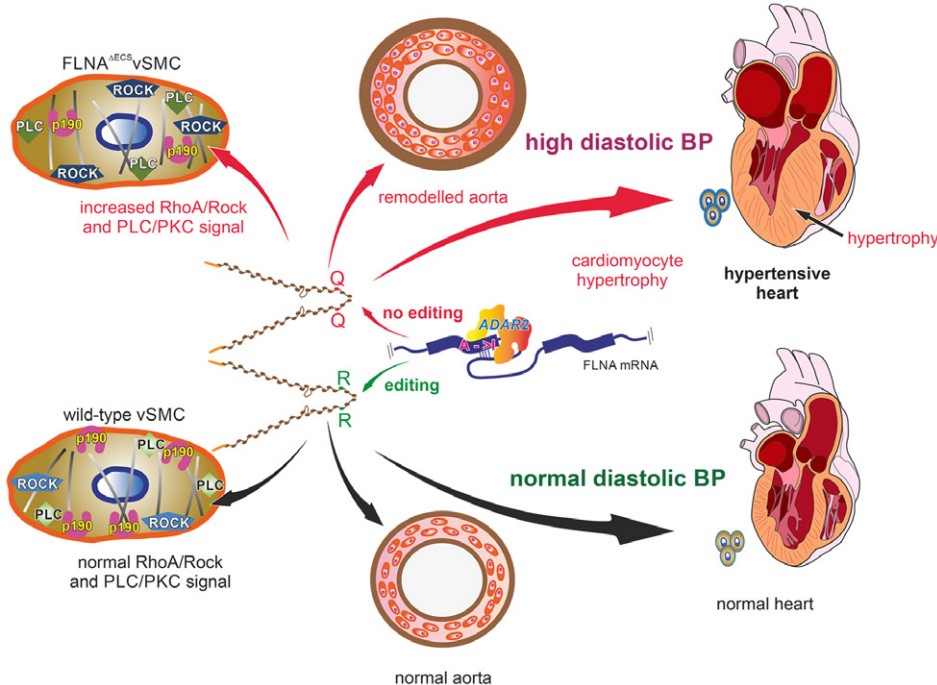

**Figure 8. Model describing consequences of Filamin A pre-mRNA editing on p190^RhoGAP localization, PLC and ROCK signaling, smooth muscle contraction, hypertension, aortic, and cardiac remodeling.**

(Top) Lack of editing in Filamin A pre-mRNA produces a Filamin A isoform that only encodes a glutamine residue (Q) at position 2431. This leads to mislocalization of p190^RhoGAP, misregulation of PLC and ROCK signaling, increased MLC phosphorylation, aortic hypercontraction, thickening of the smooth muscle layer, and increased perivascular collagen deposition. Loss of FLNA editing leads to persistently elevated diastolic blood pressure, left ventricular hypertrophy, and cardiac remodeling (Center). Filamin A pre-mRNA editing by ADAR2 triggers a Q-to-R codon exchange at the end of exon 42 (Bottom). Edited FLNA helps localize p190^RhoGAP to the cellular cortex where it can inhibit RhoA and regulate the activity of key smooth muscle contraction regulators such as PLC and ROCK machinery. This can maintain normal aortic function and normal diastolic blood pressure to preserve normal heart morphology.

Stossel *et al*, 2001; Zhou *et al*, 2010; Nakamura *et al*, 2011). The Q-to-R editing site in FLNA lies in a highly interactive region. We find that unedited FLNA can upregulate PKC and ROCK signaling, possibly through a mislocalization of p190^RhoGAP. Membrane-bound p190^RhoGAP normally promotes GTP hydrolysis by Rho GTPase, which then dampens ROCK activity. Since we showed that steady-state levels of Filamin A are unaffected by editing, we can exclude that alterations in FLNA stability contribute to our observed phenotype. With more than 100 known FLNA interacting proteins and some binding in a tension-selective manner, several interactions of FLNA are altered upon RNA editing. Globally, we observe an increased interaction of unedited FLNA^Q with the actomyosin contractile machinery. Whether an individual interaction, the crosslinking activity of Filamin, or the contractile machinery as a whole are upregulated, possibly in response to increased mechanical stress signaling, needs to be determined. Also, several genes change in their expression in the FLNA^ΔECS mice. It is obviously possible that these secondary changes may also contribute to the observed phenotype.

RNA editing is a dynamic process that responds to several internal and external conditions such as inflammation, splicing, or temperature (George & Samuel, 1999; Garrett & Rosenthal, 2012; Licht *et al*, 2016). We know that the editing status of Filamin A RNA increases dramatically during development. Future studies will

elucidate whether Filamin A RNA editing can also respond to other physiological or environmental changes.

As a remarkable decrease in FLNA editing occurs in patients suffering from cardiovascular disease, our finding that mice lacking FLNA editing also develop cardiovascular problems clearly demonstrates the biological and clinical relevance to this dominant epitranscriptomic modification. So far, only editing of one target, Azin1 RNA, was implicated in a pathology, hepatocellular carcinoma (Chen *et al*, 2013). Our work reveals a functional demonstration for editing-mediated protein recoding in the development of cardiovascular disease. Our results can stimulate the discovery of new biomarkers and therapeutic targets from a known, but overlooked, class of proteins.

# Materials and Methods

### A-to-G editing events using GTEx

The GTEx database offers high-quality RNA-Seq data of multiple tissues from hundreds of donors. We used five cardiovascular tissues available from GTEx (three arteries—aorta, tibial, and coronaries—and two heart tissues—left ventricle and left atrial appendage) and analyzed them for sites of robust A-to-G editing. We then

identified GTEx donors who suffered from a cardiovascular illness based on their medical records. We divided the cohort into a list of 210 healthy donors and 268 diseased donors, who suffered from a cardiovascular condition. We then compared these two GTEx groups to examine editing sites that significantly differentiated them.

Next, we scanned NCBI's SRA databank for high-quality, raw sequencing data from heart failure patients and recognized one major suitable cohort with data from tissue samples and three additional cohorts from peripheral blood samples: SRA ERP009437 (DCM), SRA SRP053296 (STEMI and NSTEMI), SRA SRP045355 (atherosclerosis), SRA SRP055538 (cerebral aneurism). Then, using our GTEx-derived list of candidate sites, we compared editing rates between healthy GTEx subjects and diseased specimen in each individual cohort. The main reason for using GTEx donors without cardiovascular problems as controls was that the primary pathological cohort (DCM) contained no healthy controls. Another advantage included the large volume and high quality of the GTEx cohort and the standardization it allows for.

### Identification of editing sites in the cardiovascular system

To reach a list of candidate sites edited in the different tissues in the cardiovascular system, we gradually filtered sites from the 15 million adenosines in the entire human coding sequence with slight modification for each tissue. Variations in filtering cutoffs used in stage 3 were tailored for optimal A-to-G enrichment over all other types of mismatches together with the number of predicted sites (Appendix Fig S9). Cutoffs were as follows:
**Aorta:** minimum editing, 0.05; consensus reads, 2; minimal coverage, 15; minimum supporting samples, 70.
**Ventricular Appendage:** minimum editing, 0.05; consensus reads, 2; minimal coverage, 5; minimum supporting samples, 70.
**Coronaries:** minimum editing, 0.03; consensus reads, 2; minimal coverage, 15; minimum supporting samples, 70.
**Tibial artery:** minimum editing, 0.05; consensus reads, 2; minimal coverage, 5; minimum supporting samples, 70.
**Left ventricle:** minimum editing, 0.03; consensus reads, 2; minimal coverage, 5; minimum supporting samples, 100.

Starting from ~2.5 million sites where A-to-G changes were recognized, we filtered out samples that demonstrated a high proportion of multi-editing per site (such as heavily edited A-to-G and A-to-C). Next, we filtered out sites that demonstrated significant inconsistent editing between different samples (such as significant proportion of samples showing A-to-G mismatches, while other reads were showing A-to-C mismatches). Next, we filtered for highly and reliably edited sites, which meet cutoffs for editing enrichment and consistency. Finally, we filtered out hypervariable sites, such as HLA, ribosomal genes. With this process being applied on each of the five cardiovascular tissues, a total of 252 sites in all five tissues were recognized (aorta, 92; tibial artery, 227; coronaries, 43; ventricular appendage, 68; left ventricle, 49). Out of these, 92 appear in the RADAR database (http://rnaedit.com/) and 17 of them appear in the evolutionary conserved mammalian site list. These parameters efficiently identified A-to-G editing. All A-to-G editing sites for each tissue were analyzed for average editing rate and coverage, and the pooled table for all these editing sites is represented (Table EV1). Individual tissue columns marked "yes" or "no" signify whether the site appears in the tissue.

Estimated false-positive values are given below. Version of the genome used is hg 19.

### Collection of human samples for RNA analysis

Specimens were obtained from cadavers donated to the Division of Anatomy, MUV. Written informed consent was given to use their human remains for scientific research by all body donors. All human tissue was harvested according to federal law and to the regulations of the local ethics committee under the supervision of a medical specialist in anatomy. Samples (~100 mg) were taken from the anterior tibial artery (right after its entry into the anterior compartment of the leg). Samples were either taken from donors with known cardiomyopathy or from donors free of cardiovascular disease. The cause of death was retrieved from the death certificate from each respective donor. Additionally, each heart was visually inspected thoroughly for signs or absence of cardiovascular disease. If cause of death stated in the death certificate and visual inspection were divergent, the sample would be excluded.

### Generation of Filamin A editing-deficient mice

A targeting vector was designed to delete a 228-bp-long region located in intron 42–43 harboring the editing complementary site (ECS) that base-pairs with and defines the editing site in exon 42. A long arm and a short arm consisting of regions flanking the ECS were introduced into the targeting vector. The diphtheria toxin fragment A (DTA) cassette was added after the short arm to counter-select against random integration. The ECS was replaced with a PGK-neo cassette between two loxP sites (Fig 2A). After electroporation of the construct in HM1 mouse embryonic stem cells, G418-resistant clones were verified for proper homologous recombination by Southern blotting. Next, the PGK-neo cassette was removed by transfection with pCMV-Cre. The correctly targeted clones were then injected into C57BL/6 blastocysts. Chimeric mice were backcrossed with C57BL/6 for six generations to generate isogenic lines. All animal experiments were conducted in accordance with national regulations.

### RT–PCR and determination of FLNA editing levels

Total RNA was isolated from homogenized organs with PeqGOLD TRIzol reagent (PEQLAB Biotechnologie GmbH, Germany) using the manufacturer's protocol. After DNase I treatment, cDNAs were synthesized using M-MLV Reverse Transcriptase kit (Invitrogen, Carlsbad, CA) and random hexamer primers. A FLNA cDNA fragment spanning spliced exons 42–43 was amplified from synthesized cDNAs, gel-eluted, and sequenced to check editing levels. Primers used for amplification were forward primer (5′ GTCAAGTTCAACG AGGAGCAC 3′) and reverse primer (5′ GTGCACCTTGGCATCA ATTGC 3′).

### Cell culture, immunostaining, and quantification

Vascular smooth muscle cells were derived from wt and FLNA^ΔECS aortae using the explant outgrowth method as described previously (Ray *et al*, 2001; Xu *et al*, 2009). Briefly, aortae dissected free of fat

and connective tissue were cut into small pieces and placed on gelatin-coated dishes and left undisturbed for 2 weeks for the outgrowth of vascular smooth muscle cells. The cells were then immortalized and purified from fibroblasts using a negative magnetic activated cell sorting (MACS) selection with CD90.1 microbeads (Miltenyi Biotec GmbH, Bergisch Gladbach, Germany). Purified cells were characterized by immunostaining with anti-smooth muscle alpha actin (SMA) antibodies (Appendix Fig S6; SIGMA, St Louis, MO, USA) and then checked for editing levels. Cell lines with > 90% SMA-positive population and ≥ 20% editing levels were used for further analyses. Endogenous p190$^{RhoGAP}$ localization was done in vSMCs using a p190$^{RhoGAP}$ antibody (Cell Signaling Technology, Beverly, MA), and cells were treated with 1 µM U46619 for 15 min before fixation. For quantification of p190 localization, cells were scored for their membrane vs. cytoplasmic localization and data were pooled after counting 250–300 cells in each case.

## Western blotting

Proteins were extracted using trichloroacetic acid (TCA) precipitation from primary vascular smooth muscle cells. Samples were solubilized in sample buffer containing 8 M urea and resolved on a polyacrylamide gel. Proteins were then blotted to nitrocellulose membrane and detected by MLC2 and pMLC2 (Thr18/Ser19) antibody (Cell Signaling Technology, Beverly, MA). Similarly, phosphorylation status was detected by Western blotting with either anti-phospho-MYPT1 (Thr696) polyclonal antibody (Merck, Millipore, Germany) or pCPI17 (Thr38) antibody (Santa Cruz Biotechnology, Santa Cruz, CA) on lysates of untreated cells or after treatment with either 1 µM U46619 for 15 min, 3 µM ROCK inhibitor (Y27632) for 30 min, or 5 µM PKC inhibitor (GF109203X) for 30 min. Tubulin or GAPDH was used as a loading control. All blots were detected by chemiluminescence and imaged using a CCD camera on a fusion-FX (Fisher Biotec, West Perth, WA 6005, Australia). All experiments were done at least in triplicates, and mean values ± SD were plotted.

## Rhotekin assay

RhoA activity was measured by Rhotekin assay in vSMC cells using Rhotekin RBD beads (Cytoskeleton Inc., Denver, CO) as described previously (Oinuma *et al*, 2012). Immunoblots were developed using RhoA antibody (Cell Signaling Technology, Beverly, MA).

## Aortic ring contraction assay

Aortic rings ~2 mm in length were cut from descending thoracic aortae of 16- to 20-week-old mice. The arterial rings were positioned in small wire myograph chambers (Danish Myo Technology, Aarhus, Denmark), which contained physiological salt solution (PSS; 114 mM NaCl, 4.7 mM KCl, 0.8 mM KH$_2$PO$_4$, 1.2 mM MgCl$_2$, 2.5 mM CaCl$_2$, 25 mM NaHCO$_3$, and 11 mM D-glucose pH 7.4) aerated with 5% CO$_2$/95% O$_2$ at 37°C. The myograph chambers were connected to force transducers for isometric tension recording (PowerLab; ADInstruments, Colorado Springs, MO). The rings were heated in PSS buffer to 37°C. An

initial preload of 10 mN was applied, and the rings were allowed to stabilize for 30 min. PSS containing 60 mM KCl was used to determine maximum contractility of the tissue. When the developed tension attained its peak value, the rings were relaxed by rinsing with the buffer. Next, the rings were pre-contracted with U46619 to produce 30% of the maximum contraction achieved by 60 mM KCl, followed by the addition of 3 µM Y27632, a ROCK inhibitor. Relaxation values were expressed as a percentage of the U46619 contraction. For the pre-inhibition experiments, tissues were treated with either 3 µM ROCK inhibitor (Y27632) for 30 min, 3 µM PLC inhibitor (U73122) for 20 min, or 3 µM PKC inhibitor (GF109203X) for 30 min followed by contraction using different concentrations of U46619.

## *In vitro* cell contraction assay

xCELLigence RTCA system (Roche Applied Science, Mannheim, Germany) was used to measure vSMC contraction as described previously (Wang *et al*, 2013). Briefly, wt and FLNA$^{ΔECS}$ vSMCs were seeded at 25,000 cells/well in 96-well E-plates and monitored for attachment and growth for the next 24 h. Subsequently, cells were treated with different U46619 concentrations (0.01–10 µM), and cell impedance or cell index (CI) was live-monitored for the next 2 h. For data analyses, CI was normalized to 1 at the time of addition of the agonist and minimum CI value (CI$_{min}$) within 30 min of agonist addition was used to calculate the percentage of contraction.

## Blood pressure measurements

Sixteen- to 20-week-old male mice were used to measure blood pressure by radiotelemetry in a blinded setup. Blood pressure, heart rate, and activity were continuously recorded for 72 h as described (Huetteman & Bogie, 2009). Data were sampled in 1-min intervals. In parallel, an activity profile was collected allowing to correlate active and inactive phases with systolic and diastolic blood pressure values. Data collection was performed using the Ponemah software (DSI, IL, USA).

## Surgical implantation of blood pressure and ECG telemetry transmitters

HDX-11 telemetry transmitters (Data Sciences International—DSI, USA) for blood pressure and ECG recordings were implanted subcutaneously using the following procedure: Anesthesia was induced using 4% isoflurane and maintained at 1.5–2% isoflurane at a flow of 1–2 l/min. The gel-filled blood pressure catheter was placed in the left carotid artery and positioned so that the sensing region of the catheter was in the aortic arch. The transmitter portion of the device was positioned along the lateral flank between the forelimb and hindlimb. A subcutaneous pocket was formed by blunt dissection from the neck incision down along the animal's flank. The biopotential leads for ECG measurements were routed subcutaneously from a small neck incision, so that the positive lead was positioned ~1 cm left of the xiphoid process and the negative lead was positioned at the right pectoral muscle. Mice were allowed to recover in their home cages for at least 11 days and treated with 0.027 mg Rimadyl/ml drinking water (analgesic) and 0.17 mg

Baytril/ml drinking water (antibiotic) during the recovery period. After recovery, telemetric measurements were taken by placing the mice in their homecages on RPC-1 receiver plates (DSI) to detect the radio signal emitted from the HDX-11 transmitters that was converted by DSI data exchange matrices and analyzed with Ponemah v5.20 software (DSI).

## Mass spectrometry

50 μl/IP of protein A beads was incubated with 50 μl/IP of polyclonal α-FLNA antiserum for 1 h at 4°C. After incubation, beads were placed onto a magnetic rack and washed with lysis buffer and sodium borate (0.2 M, pH = 9). The antibody was crosslinked in 20 mM dimethyl pimelimidate (DMP) in 0.2 M sodium borate, for 30 min at room temperature. Crosslinking was followed by three washing steps (5–10 min) with 250 mM Tris (pH = 8.0) and a quick pre-elution step with 100 mM glycine (pH = 2.0). Beads were washed and stored in 1xPBS prior to usage. Crosslinked beads were aliquoted and added in equal amounts to the cell lysates. Beads were incubated for 1 h at 4°C on a rotating wheel and washed with 6 × 500 μl/IP of wash buffer. A small aliquot (10%) was taken for IP control and the rest submitted to mass spectrometry for further processing. Detected proteins were analyzed and quantified by Perseus algorithm (Tyanova *et al*, 2016) and tested for statistically significant differences from wild-type and FLNA editing-deficient cells. Data were analyzed after normalizing FLNA intensities in all the samples.

## Histology and morphometric measurements

Dorsal aortae were dissected from 5- to 6-month-old mice and fixed with 4% paraformaldehyde for 3 h at RT and processed for cryosectioning. 3 μm cryosections were taken using cryostat (HM 500 OM; Microm, Walldorf, Germany) and stained with either hematoxylin–eosin or Elastica van Gieson staining kit (Merck Eurolab, Darmstadt, Germany). The luminal diameter, media area, and adventitial area were analyzed using ImageJ software. For the overall measurements, four sections were randomly chosen from each sample and four mice were analyzed and averaged.

Hearts were excised from 5- to 6-month-old mice and fixed with 4% paraformaldehyde in PBS overnight at 4°C and processed for cryosectioning. 3 μm cryosections were taken using cryostat (HM 500 OM; Microm, Walldorf, Germany) and stained with Masson's trichrome kit (RAL Diagnostics, Martillac, France) to calculate perivascular fibrosis. Images were processed with ImageJ software to assess perivascular fibrosis, which was calculated as the ratio of fibrosis area surrounding the vessel to the total vessel area. Five sections were examined in each heart, and results were obtained from the average of four hearts in each group.

To measure the lumen and wall thickness of resistance vessels, kidney paraffin sections were stained with α-SMA (alpha smooth muscle actin) antibody and the lumen (inner area) and the wall thickness (outer radius-inner radius) were calculated. To address the thickness of heart walls, 6-month-old mice were sacrificed, and their hearts were perfused with 4% paraformaldehyde in PBS prior to dissection. Hearts were further fixed overnight at 4°C, washed 3 × 20 min with PBS, and processed for paraffin embedding. Samples were sectioned at 5 μm thickness and stained with hematoxylin–eosin. Six sections from each heart were analyzed with ImageJ software, and results were represented as average heart wall thickness of either interventricular wall septum or left lateral ventricular wall.

Additional sections from the same hearts were deparaffinized, hydrated, and stained with WGA (wheat germ agglutinin, Sigma-Aldrich) for 1 h at room temperature to visualize cell membranes. Respective images were analyzed by ImageJ software to calculate average cardiomyocyte cross-sectional area. Results were obtained from six mice per genotype.

## Statistical analysis

Data were analyzed using Student's *t*-test with equal or unequal variance assumptions and two-way ANOVA followed by a Bonferroni *post hoc* test for myography experiments (using GraphPad Prism 5.0). Data sets failing normality were analyzed using nonparametric comparisons (Mann–Whitney *U*-test). $P < 0.05$ was considered to be statistically significant.

## MRI data acquisition

Magnetic resonance imaging (MRI) was performed on 15.2 T Bruker Biospec (Ettlingen, Germany) using a 35-mm volume birdcage coil. Mice were anesthetized with isoflurane (3% induction, 1–1.5% maintenance). Heart rate was monitored during each study using gold disk surface electrodes attached to the front and hind paws (SA Instruments Inc., Stony Brook, NY). A special care was taken to keep heart rate as constant as possible (mean 461 ± 12 beats/min), with appropriate anesthesia adjustments during imaging studies.

Initially, several 2D fast low-angle shot (FLASH) scout images were recorded in the transverse and sagittal plane to aid localization of the heart and aorta. Short-axis cardiac images were acquired by positioning slices orthogonal to the long-axis cross-sections of the heart. To keep sampling rate ≥ 130 Hz, each slice of the cardiac short axis was acquired separately. A total of 8 ± 1 short-axis slices were acquired for each mouse, providing complete coverage of the left ventricle. For each short-axis slice, a series of 12 images were acquired using ECG-gated cine FLASH sequence (repetition time (TR), 10–11 ms depending on the R-R interval; echo time (TE), 1.5 ms; FOV, $2.5 \times 2.5$ cm$^2$; matrix size, $256 \times 256$, 1 mm slice thickness, 6 averages). Next, images of the aorta were acquired using same parameters and cine FLASH sequences. Slices were positioned perpendicular to the aorta immediately before brachiocephalic trunk and through the aortic arch to account for spatial displacement of the aorta during the cardiac cycle. To acquire quantitative velocity maps of the aorta, velocity encoding (VENC) sequence was used with TR/TE, 6/2.5 ms, 20 frames; FOV, $2.5 \times 2.5$; matrix size, $256 \times 256$, 1 mm slice thickness, 6 averages; and maximum blood flow velocity, 240 cm/s. The velocity profiles were measured in one direction, parallel to the flow and orthogonal to the aorta cross-sectional plane. The imaging slice was positioned just above the aortic valve for $N = 4$ wt and $N = 5$ FLNA$^{\Delta ECS}$ mice. To correlate velocity profiles with anatomical location and correct for aorta movement during the cardiac cycle, FLASH sequence with same slice location and parameters as VENC sequence was acquired.

## Data analysis

End-diastolic and end-systolic right and left ventricular volumes were estimated from short-axis cardiac images. National Institute of health (NIH) software ImageJ (http://imagej.nih.gov/ij/) was used for data analysis. For all slices, end-systolic and end-diastolic volumes were identified by visually inspecting all 12 frames for filling size of all chambers as well as wall thickening patterns. In all experiments, the first cine frame after triggering on the QRS was the frame with maximal ventricular area and was referred as end dia-stole. All volume estimates were done by the investigator blinded to mouse genotype. For the circumferential strain measurement, the time course of the aorta cross-sectional area $A(t)$ and vessel wall radius $(r(t))$ were estimated during systole and diastole. The follow-ing was assumed: The deformation through the thickness of the vessel wall was negligible, and the deformation in the axial direction was small compared to the circumferential deformation. The circumferential cyclic strain was calculated in two ways: (i) from images perpendicular to the aorta, using cross-sectional area of the aorta (Morrison *et al*, 2009), and (ii) from the images through the aortic arch, using the diameter of aorta and assuming a circular cross-section of the aorta. In both cases, the following equation was used (Herold *et al*, 2009):

$$E_{\theta\theta}(t) = \frac{1}{2}\left[\left(\frac{r(t)}{r(0)}\right)^2 - 1\right] = \left[\frac{A(t)}{A(0)} - 1\right]\Big/2.$$

VENC protocol allowed quantitative maps in [cm/s] of the blood flow through the aorta. Aortic velocity waveform, representing mean velocity during different stages of cardiac cycle, is shown in Fig 7.

Data were first tested for normal distribution and were thereafter analyzed using *t*-test using Sigma Plot 13 (Jandel Scientific Software, GmbH).

## Data availability

Mass spectrometry data of FLNA[Q] and FLNA[R] interacting proteins are deposited at PRIDE: https://www.ebi.ac.uk/pride/archive/projects/PXD009769

**Expanded View** for this article is available online.

## Acknowledgements

The authors would like to thank Lena Hirtler for supervising human tissue isolation. We thank the VBCF preclinical phenotyping facility, Sylvia Badurek, and Mumna Al Banchaabouchi for mouse blood pressure measurement, Jelena Zinnanti for MRI imaging, and the MFPL mass spectrometry facility for mass spectrometry analysis. Irmgard Fischer and Ingrid Hammer are thanked for help with sectioning and staining of mouse tissues. Michael Janisiw and Peter Burg are thanked for their excellent technical assistance. We thank Thomas Nardelli for help with preparation with graphics. Margarete Lechleitner is acknowledged for help with myography experiments. The German Mouse Clinic team is thanked for thorough phenotyping of mice. We thank Life Science Editors for technical, editing assistance. This work was supported through grant numbers F4313, P22956, and P27166-B23 of the Austrian Science Foundation to MFJ and SF, respectively. M. Stulic was supported through the Austrian Science Foundation Doctoral Program W1207. LC was supported by the Mahlke-Obermann Stiftung and the European Union's Seventh Framework Programme (FP7) Marie Curie Actions (Grant Agreement No. 609431)/INDICAR —Interdisciplinary Cancer Research. M. Sibilia was funded by the EC programs QLG1-CT-2001-00869 and LSHC-CT-2006-037731 (Growthstop), the Austrian Federal Government's GEN-AU program "Austromouse" (GZ 200.147/1-VI/1a/2006), and the Austrian Science Fund grants FWF-P18421, FWF-P18782, SFB-23-B13, and DK W1212. EYL was supported by the European Research Council (311257) and the Israel Science Foundation (1380/14). In addition, the study was partially funded by the German Federal Ministry of Education and Research to the GMC (Infrafrontier Grant 01KX1012).

## Author contributions

MJ, MSt, SPR, AK, DP, XS, TK-R, VG-D, HF, MHA, JZ, LC, DM, and RB, performed experiments. CR and LR provided human samples. TDM and EYL performed bioinformatics analysis. EP performed statistical analysis. MJ, MSt, SPR, DP, KM, WFG, MSi, SF, and MFJ planned experiments. MJ, MSt, KM, SF, and MFJ evaluated data. MJ and MFJ wrote the manuscript.

## Conflict of interest

The authors declare that they have no conflict of interest.

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
