## [Review Process File · The EMBO Journal]

RNA editing of filamin A pre-mRNA regulates vascular contraction and diastolic blood pressure

Mamta Jain, Tomer D. Mann, Maja Stulić, Shailaja P. Rao, Andrijana Kirsch, Dieter Pullirsch, Xué Strobl, Claus Rath, Lukas Reissig, Kristin Moreth, Tanja Klein-Rodewald, Raffi Bekerdjian, Valerie Gailus-Durner, Helmut Fuchs, Martin Hrabě de Angelis, Eleonore Pablik, Laura Cimatti, David Martin, Jelena Zinnanti, Wolfgang F. Graier, Maria Sibilía, Saša Frank, Erez Y. Levanon and Michael F. Jantsch.

Review timeline:

Submission date:	19 th May 2016
Editorial Decision:	25 th July 2016
Additional correspondence	17 th November 2017
Additional correspondence	22 nd November 2017
Revision received:	24 th November 2017
Additional correspondence	15 th January 2018
Editorial Decision:	15 th March 2018
Revision received:	18 th May 2018
Editorial Decision:	22 nd June 2018
Revision received:	25 th June 2018
Accepted:	27 th June 2018

Editor: Anne Nielsen

Transaction Report:

1st Editorial Decision

25th July 2016

Thank you for submitting your manuscript for consideration by The EMBO Journal and my apologies for the extended duration of the review period in this case. Your study was sent to three referees and we have now received their comments (included below). As you will see, the referees all express interest in the work and topic in principle, but unfortunately they all do not offer strong support for publication in The EMBO Journal.

While ref #2 is very positive about the study and highlights that your finding of a physiological contribution from a single RNA-editing site outside the CNS will be a valuable starting point for future work, refs #1 and #3 raise concerns about the strength of the effect seen and the mechanism involved, respectively. Consequently, neither of these two refs support publication in The EMBO Journal in their recommendations to the editorial office. Clearly, an extensive amount of further experimentation would be required to address the issues raised by the referees and to bring the study to the level of insight and significance required for publication here.

However, at the same time it is clear that the referees find the study to be mostly convincingly done for the available data and I have therefore taken the liberty to discuss the manuscript with my colleague Esther Schnapp at our sister journal EMBO Reports (she also saw you present the study at the Epigenomics meeting at EMBL in April). The outcome is that Esther would be happy to offer publication of your study in EMBO Reports if you were to submit a revised manuscript there (you should address/comment on the major and minor points of ref #1 but you will not have to look for effects in other tissues). In that way your manuscript will not have to go to new referees at EMBO

Reports but will be handled based on the comments from our referees.

Thank you in any case for the opportunity to consider this manuscript. I am sorry we cannot be more positive for The EMBO Journal on this occasion, but I hope that you will take the chance to submit a revised version of this manuscript to EMBO Reports. Feel free to contact either Esther or me with any questions about the revised manuscript.

REFEREE REPORTS.

Referee #1:

This is an interesting paper that demonstrates a novel mechanism of blood pressure control by RNA-editing of filamin A pre-mRNA. The authors have shown that lack of filamin A pre-mRNA editing leads to increased stress fiber density and elevated myosin light chain phosphorylation, which could explain the phenotype. Although the topic of this paper is interesting and may satisfy researchers in the field, there are some major and minor concerns that the authors should address in order to be accepted for publication.

Major concerns

1. Although the authors mention that the diastolic BP has increased "clearly" in the FLNA Δ ECS mice, I have to say that the difference is trivial. I think another 2 or 3 mice should be added to each group to clarify the diastolic BP difference between the 2 groups.
2. The authors explain that FLNA Δ ECS mice have high diastolic BP which eventually develop heart failure. To explain that the cardiomyocyte hypertrophy and perivascular fibrosis are not due to independent effects by lack of RNA-editing, histological analysis should be shown in younger mice such as 3-4 weeks or even younger, before getting influenced by the high blood pressure. Moreover, the histological analysis has been done in 5-6 month old mice, while the heart function analysis has been done in 24 month aged mice, which is a very limited situation. According to histological analysis of 5-6 months old mice, they could also have signs of heart failure. Why does it take so long to develop heart failure when the 5-6 months old mice already show histological signs of hypertensive heart disease? Also it would be interesting to histologically examine the effect on other organs sensitive to hypertension, for example the brain or the kidney, in order to explain that hypertension is causing organ damage.

Minor concerns

1. Although the authors have shown MRI data of the heart, a macro picture of the heart of WT and FLNA Δ ECS mice, showing enlargement in FLNA Δ ECS mice should be presented. Also a cross section of the heart showing hypertrophy of the walls should be presented.
2. The quality of the WB picture in Fig4D is low. A better quality picture should be shown.
3. Page 13 line 25
" we can also exclude that an alterations in..."
" we can also exclude that alterations in..."

Referee #2:

Only a handful of mammalian re-coding ADAR editing sites have been characterized sufficiently to understand the mechanistic effects of re-coding. Excitingly, this paper provides an extensive and careful characterization of a re-coding event in filamin A (FLNA), a member of the filamin family of actin-crosslinking proteins. The studies are centered on a mouse the authors generated that is exclusively deficient in FLNA editing (FLNA Δ ECS), although FLNA protein levels are normal. Starting with the observation that the mice exhibit increased contraction of smooth muscle, the authors use histological, molecular and physiological assays to convincingly delineate the pathway affected, and the physiological consequences of a lack of the single editing site in FLNA, which

leads to aortic hypercontraction, hypertension and high blood pressure. The data are extremely nice, the experiments are very well controlled, and statistical significance is established for all experiments. Important conclusions are summarized below.

- Using multiple assays the authors show that mice lacking FLNA RNA editing show increased contraction of smooth muscle cells and elevated stress fiber density.
- Consistent with the altered contraction, factors known to be involved in the control of smooth muscle contraction are altered in the absence of FLNA editing. Phosphorylation of myosin light chain is increased, activation of RhoAGTP is elevated, and myosin light chain phosphatase 1 is inhibited by increased phosphorylation.
- As a first step in bringing these observations to the molecular level, the authors' show that unedited FLNA leads to mislocalization of p190RhoGAP. In the presence of edited FLNA, p190RhoGAP properly localized to the cell cortex, while without editing it was dispersed throughout the cytoplasm. Addition of a contraction activator caused p190RhoGAP to disperse in the cytoplasm in the presence of edited FLNA, while localization was unaffected in cells expressing the unedited FLNA.
- FLNA^{ΔECS} mice are hypertensive, especially during resting and sleeping phases, indicating that FLNA editing is important for lowering of the diastolic blood pressure during phases of relaxation.
- Aorta of FLNA^{ΔECS} mice showed increased lumen diameter and a thickening of the medial wall, accompanied by increased collagen deposition.
- FLNA^{ΔECS} mice showed reduced cardiac output, a hallmark of heart failure.

As mentioned, the data are very convincing, and all experiments include multiple biological and technical replicates. The manuscript will be interesting to scientists interested in editing as well as the many scientists focused on understanding elevated diastolic blood pressure and subsequent heart failure. While additional molecular details will no doubt be forthcoming, the data lay the foundation for these studies, and significantly advance the field.

I have only a couple of minor comments:

1. I am a bit confused by the sequencing chromatograms shown in 1B. Does the sequence below the WT colon have an extra G in the text? (TTCAGGGA)
2. KPSS should be defined in Figure 2.

Referee #3:

The authors have generated mice with a point mutation that abrogates FLNA editing. This mutation results in altered signaling and contractile properties of smooth muscle cells, presumably leading to an observed hypertension accompanied by perivascular fibrosis and myocyte hypertrophy. The results are overall convincing. Some of the wording needs to be altered, including statements such as "To understand the impact of RNA editing on the contractile apparatus and actin organization", which implies a general role in RNA editing rather than the specific role of FLNA editing. Importantly, the authors have explored some aspects of FLNA biology, but the actual function of the RNA editing is not apparent, so the study remains anecdotal.

Additional correspondence (authors)

17th November 2017

I am contacting you to enquire whether you would consider an entirely reorganized and expanded version of our previously submitted manuscript for EMBO J.

As a quick background: In the version we had submitted earlier last year we had shown that lack of RNA editing of the Filamin pre-mRNA leads to increased contraction of smooth muscle cells by upregulation of the entire smooth muscle contractile signaling landscape. We had also shown that this- in the long run -would lead to elevated blood pressure and to cardiac problems in mice.

The reviewers back then had several comments that could (and were meanwhile) addressed.

However, in the meantime we have teamed up with the group of Erez Levanon who could show that

filamin editing is the major editing change found associated with cardiovascular disease in humans. We could also show that Filamin is so massively edited that it may be the single most edited transcript in the human transcriptome, leading to a protein recoding event.

As I meanwhile work at the medical university we also looked at the editing status of dorsal and tibial aorta of human cadavers. Here we find a strong correlation between a decrease in FLNA editing and the development of cardiomyopathy.

We also performed proteomics of immunoprecipitations from smooth muscle cells of wild type and FLNA editing deficient cells. This data shows a differential interaction of FLNA with proteins involved in cellular contraction and that these interactions are modulated by RNA editing.

Taken together, the manuscript has shifted its focus strongly and now includes a strong point demonstrating the link between FLNA editing and the development of cardiovascular disease in humans. By combining this data with the analysis of a transgenic mouse deficient in FLNA editing, we can prove the functional link between editing and disease development, that goes far beyond a simple correlation.

Also on the animal analysis side we have added more blood pressure data measurements, and more contraction studies showing that role of Ca²⁺ release in differential contraction.

I would now like to enquire whether you would consider this version of the manuscript. As the focus of the manuscript has strongly shifted towards a human transcriptome analysis of GTEx data, I would consider it more suitable to see this manuscript as a potential new submission. However, should you consider our manuscript, I would of course follow your suggestion.

Additional correspondence (editor)

22nd November 2017

Thank you for contacting us about the new version of your manuscript. I have now read the new version and looked at the referee concerns from the last round. The conclusion is that we would be interested in considering the revised version and I would therefore encourage you to formally submit your study here using the link provided below.

1st Revision - authors' response

24th November 2017

Rebuttal Letter

We are thankful for the helpful comments. We have tried to address them wherever feasible. The detailed point to point replies are listed below.

Importantly, however, we have shifted the focus of the manuscript and performed many additional experiments since the first submission. The new version starts with a bioinformatic analysis of human RNA seq data derived from GTEx data and human cardiovascular disease data. This analysis shows that RNA editing of FLNA is critically lowered in disease condition. Moreover, we show that editing of filamin A pre-mRNA is the most abundant editing event in the human body.

A mouse model in which FLNA editing has been impaired proves that lack of FLNA editing impacts on cardiovascular health and establishes a causal relationship between filamin A editing levels and cardiovascular health.

Point by point response to previous comments raised by reviewers :

Referee #1:

This is an interesting paper that demonstrates a novel mechanism of blood pressure control by RNA-editing of filamin A pre-mRNA. The authors have shown that lack of filamin A pre-mRNA editing leads to increased stress fiber density and elevated myosin light chain phosphorylation, which could explain the phenotype. Although the topic of this paper is interesting and may satisfy researchers in the field, there are some major and minor concerns that the authors should address in order to be accepted for publication.

Major concerns

1. Although the authors mention that the diastolic BP has increased "clearly" in the FLNA Δ ECS mice, I have to say that the difference is trivial. I think another 2 or 3 mice should be added to each group to clarify the diastolic BP difference between the 2 groups.

We thank the reviewer for a valid argument. We have now included more mice and have now compared 9 wild type and 10 mutant mice. Our original claim that diastolic blood pressure is elevated in mutant mice could be substantiated in the new data set.

2. The authors explain that FLNA Δ ECS mice have high diastolic BP which eventually develop heart failure. To explain that the cardiomyocyte hypertrophy and perivascular fibrosis are not due to independent effects by lack of RNA-editing, histological analysis should be shown in younger mice such as 3-4 weeks or even younger, before getting influenced by the high blood pressure.

We are thankful for this comment and we have now checked the cardiomyocyte size and perivascular fibrosis in young mice (3-4 weeks) and found no difference in both the parameters (see supplementary Fig S10). We have previously shown that FLNA editing is low in young mice and starts to get highly edited after 3 months of age. Hence, the effects of loss of FLNA editing are not visible at such an early stage and they start to mount once FLNA editing is increasing and are quite visible by the age of 5-6 months.

Moreover, the histological analysis has been done in 5-6 month old mice, while the heart function analysis has been done in 24 month aged mice, which is a very limited situation. According to histological analysis of 5-6 months old mice, they could also have signs of heart failure. Why does it take so long to develop heart failure when the 5-6 months old mice already show histological signs of hypertensive heart disease?

We appreciate this concern. Indeed we performed MRI measurements on the same set of mice at one year of age but did not find significant signs of heart failure. Apparently, development of the heart failure is a late event in our mice that is also not a 100% prominent. For the same reason, we have not focused much on these observations and have kept our claims more towards FLNA editing role on hypercontraction and their effects on blood pressure regulation.

Also it would be interesting to histologically examine the effect on other organs sensitive to hypertension, for example the brain or the kidney, in order to explain that hypertension is causing organ damage.

We thank the reviewer for this comment. However, at this stage we believe that these experiments are beyond the scope of the manuscript at this stage. In order to do them properly, a larger cohort of mice would need to be aged, fixed, sectioned, and analyzed at several ages. We will consider such analysis for the

future.

Minor concerns

1. Although the authors have shown MRI data of the heart, a macro picture of the heart of WT and FLNA Δ ECS mice, showing enlargement in FLNA Δ ECS mice should be presented. Also a cross section of the heart showing hypertrophy of the walls should be presented.

We have now included the sections of the heart clearly showing the left ventricular wall thickening in FLNA Δ ECS mice (Fig.6E).

2. The quality of the WB picture in Fig4D is low. A better quality picture should be shown.

We have now replaced the western blot picture.

3. Page 13 line 25

" we can also exclude that an alterations in..."

" we can also exclude that alterations in..."

We thank the reviewer for pointing out this grammatical error, it has been corrected in the current version.

Referee #2:

Only a handful of mammalian re-coding ADAR editing sites have been characterized sufficiently to understand the mechanistic effects of re-coding. Excitingly, this paper provides an extensive and careful characterization of a re-coding event in filamin A (FLNA), a member of the filamin family of actin-crosslinking proteins. The studies are centered on a mouse the authors generated that is exclusively deficient in FLNA editing (FLNA Δ ECS), although FLNA protein levels are normal. Starting with the observation that the mice exhibit increased contraction of smooth muscle, the authors use histological, molecular and physiological assays to convincingly delineate the pathway affected, and the physiological consequences of a lack of the single editing site in FLNA, which leads to aortic hypercontraction, hypertension and high blood pressure. The data are extremely nice, the experiments are very well controlled, and statistical significance is established for all experiments. Important conclusions are summarized below.

- Using multiple assays the authors show that mice lacking FLNA RNA editing show increased contraction of smooth muscle cells and elevated stress fiber density.
- Consistent with the altered contraction, factors known to be involved in the control of smooth muscle contraction are altered in the absence of FLNA editing. Phosphorylation of myosin light chain is increased, activation of RhoAGTP is elevated, and myosin light chain phosphatase 1 is inhibited by increased phosphorylation.
- As a first step in bringing these observations to the molecular level, the authors' show that unedited FLNA leads to mislocalization of p190RhoGAP. In the presence of edited FLNA, p190RhoGAP properly localized to the cell cortex, while without editing it was dispersed throughout the cytoplasm. Addition of a contraction activator caused p190RhoGAP to disperse in the cytoplasm in the presence of edited FLNA, while localization was unaffected in cells expressing the unedited FLNA.
- FLNA Δ ECS mice are hypertensive, especially during resting and sleeping phases, indicating that FLNA editing is important for lowering of the diastolic blood pressure during phases of relaxation.
- Aorta of FLNA Δ ECS mice showed increased lumen diameter and a thickening of

the medial wall, accompanied by increased collagen deposition.

- FLNA Δ ECS mice showed reduced cardiac output, a hallmark of heart failure.

As mentioned, the data are very convincing, and all experiments include multiple biological and technical replicates. The manuscript will be interesting to scientists interested in editing as well as the many scientists focused on understanding elevated diastolic blood pressure and subsequent heart failure. While additional molecular details will no doubt be forthcoming, the data lay the foundation for these studies, and significantly advance the field.

I have only a couple of minor comments:

1. I am a bit confused by the sequencing chromatograms shown in 1B. Does the sequence below the WT colon have an extra G in the text? (TTCAGGGA)

We thank the reviewer for pointing out the mistake, it should be TTCGGGA. This has now been corrected in the sequencing chromatogram.

2. KPSS should be defined in Figure 2.

This has been included in the methods section.

Referee #3:

The authors have generated mice with a point mutation that abrogates FLNA editing. This mutation results in altered signaling and contractile properties of smooth muscle cells, presumably leading to an observed hypertension accompanied by perivascular fibrosis and myocyte hypertrophy. The results are overall convincing. Some of the wording needs to be altered, including statements such as "To understand the impact of RNA editing on the contractile apparatus and actin organization", which implies a general role in RNA editing rather than the specific role of FLNA editing. Importantly, the authors have explored some aspects of FLNA biology, but the actual function of the RNA editing is not apparent, so the study remains anecdotal.

We thank the reviewer for raising a justified concern. In order to understand the mechanism behind the role of FLNA editing in cellular contraction, we have now done a mass spec analysis of proteins associated with editable and unedited FLNA immunoprecipitated from vSMC cells derived from WT and FLNA Δ ECS mice. We show that unedited FLNA binds more strongly to many proteins involved in the contractile machinery. Clearly, not a single factor is causing an increased contractility. We have several ongoing experiments addressing this point and hope to be able to document this point in a clearer fashion in the future.

Additional correspondence

15th January 2018

Thank you for submitting your revised manuscript to The EMBO Journal and my apologies for the extended duration of the review period over the holidays. Your study has now been seen by two referees (one original and one new) and their comments are included below.

As you will see from the reports, referee #1 (same as ref #1 from the first round) is largely satisfied with the clarifications that have been provided in the new version of the manuscript. On the other hand, Ref #2 (new referee) is much more critical and finds that both the mechanistic and functional aspects of the study would have to be extended substantially in order for the manuscript to be a strong candidate for publication in The EMBO Journal. I realise that some of these points are rather further reaching and may fall outside the scope of a revision but it's clear that more data will have to be included before we can take further steps towards publication. At this stage - and given the previous offer from EMBO Reports to publish a revised version of the manuscript there - I would

therefore like to discuss the experiments that could be included in another potential round of revision before I go on to make an official decision on this manuscript.

I would ask you to take a look at the reports included below and let me know what kind of data you would be able - and willing - to include in a potential revision to address the concerns from referee #2 (both in terms of controls for conclusiveness and for further functional insight). I would then take that into consideration for the final decision on your study. The aim of this is ultimately to prevent you from working extensively on a revision that would have little chance of convincing the referees - and also to find a solution that works for you in terms of additional time investment in this long-running project.

You can send me the outline for a possible revision (or a preliminary point-by-point response) and I will then get back to you with a decision.

2nd Editorial Decision

15th March 2018

Thank you again for submitting a revised version of your manuscript as well as a preliminary point by point response to the concerns raised by the new referee. I would also like to thank you for your patience with the extended duration of the re-review period as well as the subsequent discussions.

As I mentioned in my last letter, you will see that referee #1 supports publication while ref #4 (new referee) raises a number of concerns about the mouse model, human data and mechanism at play. I realise that several of these points may fall outside the scope of the current study and after reading your response to the full list of concerns - and discussing it with my colleagues in the editorial team - I would like to invite you to submit a final revision of your manuscript to The EMBO Journal, along the lines outlined in your response.

While many of the concerns raised by ref #4 can be addressed with text revisions/clarifications, I would suggest that you include the additional data for ADAR2 expression in patient tissue and resistance arteriole thickness in FLNA-DEC mice that you mention in the response to ref #4.

 REFEREE REPORTS.

Referee #1:

This manuscript demonstrates a novel contribution of RNA-editing, specifically in filamin A pre-mRNA in controlling blood pressure. The authors have shown that lack of filamin A pre-mRNA editing leads to increased phosphorylation of myosin light chain and an increase in RhoA/Rock and PLC/PKC signaling, which could explain the increased smooth muscle contraction in FLNA Δ ECS vSMCs. The experiments show convincing data about the in vivo vasculopathy in FLNA Δ ECS mice and mass spec analysis shows that unedited FLNA binds stronger to proteins involved in the contractile machinery, which could be one of the mechanisms. It is interesting that the human GTEX data set showing decreased RNA editing was analyzed from dilated cardiomyopathy patients where the pathology is a primary cardiomyopathy (cardiomyocyte disease), but the FLNA Δ ECS mice show only secondary cardiomyocyte hypertrophy and the phenotype is rather in smooth muscle cells, demonstrating diastolic hypertension. Indeed smooth muscle cells in the heart could be contributing to the phenotype in humans or it could be another mechanism or just the difference between species. I think the manuscript will be interesting enough to investigators studying smooth muscle cell biology and vasculopathy. I just have a minor concern about the figure.

Minor concern

1. Fig7A is a little confusing to me. The number of WT mice are explained as 4 in the manuscript but the figure seems to have 5 solid lines representing WT. Also I do not see a pink solid line for an outlier, it rather looks purple to me if I am understanding the figure correctly.

Referee #4:

The submitted revised manuscript entitled "RNA editing of Filamin A pre-mRNA regulates diastolic blood pressure and cardiovascular remodeling" by Jain M, et al. describes the extent of RNA editing of Filamin A in various human tissues ranging between 8 and 92% based on an RNA editing analysis of the Genotype-Tissue Expression project (GTEx) as well as in human tissue specimens from the tibial artery and the aorta of control subjects and patients with a positive history of a cardiovascular disease. The authors report a decrease of about 15-20% of the extent of RNA editing rate of Filamin A in aortic-arterial tissues from subjects with a positive history of cardiovascular disease. Mechanistically the authors created a transgenic mouse strain by deleting a 228 bp long intronic region located in intron 42-43 which works as the editing complementary site (ECS) of the double-stranded RNA formed by this region and the exon 42. It is known that RNA editing of filamin A pre-mRNA at a specific adenosine in the exon 42 leads to a Q/R substitution in Ig-repeat 22 of the encoded protein. The authors show that the deletion of the 228bp intronic ECS results in absence of RNA editing of the specific exon 42 adenosine. The aortic ring contraction from mice deficient for the Filamin A ECS was found to be up to 25% increased after treatment with the thromboxane A2 receptor agonist U46619 compared to WT aortic rings. Mouse studies utilizing only 16-24 week old male mice revealed a slight increase of around 5-8 mmHg the diastolic arterial pressure in the resting, but not in active phase. Histological analysis of the aorta cross-sections revealed a thicker adventitial area in the Filamin A transgenic mice. Heart sections showed an increase in perivascular fibrosis and a significant increase in the left ventricular wall thickness as well as of the cardiomyocyte cross-sectional size. Mechanistically, the authors show that deletion of the Filamin A ECS in murine primary aortic smooth muscle cells induces the phosphorylation of myosin light chain, the levels of RhoA-GTP, the phosphorylation of myosin light chain phosphatase 1 (MYPT1) and of CPI-17, which are all critical factors for smooth muscle cell contraction. Inhibition of either the ROCK or the PKC signaling abolished the CPI-17 effect. Further, the authors show that the localization of p190-RhoGAP is affected. Interestingly, the authors performed a Filamin A immunoprecipitation followed by mass spectrometry in three technical replicates showing that from the 300 proteins, 20 were enriched in the mutant (unedited) Filamin A, while 30 were enriched in the WT (edited) Filamin A form. Last, the authors performed cardiac MRI showing that the velocity in the ascending aorta was decreased in the Filamin A transgenic mice during the systolic phase.

Although the authors should be admittedly congratulated for the extent of their efforts to analyse in depth the effect of RNA editing of Filamin A in the cardiovascular homeostasis, the study unfortunately lacks scientific coherence and cohesion and the presented very interesting findings are poorly connected with each other. Further, the deletion of a 228 bp intronic area instead of the specific replacement of only the nucleotide of interest responsible for the amino acid exchange (Q2341R) limits the interpretation and relevance of the results. The revised manuscript does not fully address the concerns of the Reviewer 1 related to the end-organ effects of arterial hypertension and of the Reviewer 3 related to the absence of a mechanism related to RNA editing that connects the described phenotypes. The dynamic landscape of RNA editing of repetitive elements and coding regions including Filamin A in human tissues of the GTEx has been recently reported (Tan MH et al., *Nature*. 2017 Oct 11;550(7675):249-254). The role of Filamin A in smooth muscle cell function has been also recently reported (Retailleau K et al., *Cell Rep*. 2016 Mar 8;14(9):2050-2058) showing a distinct differential role of Filamin A in basal blood pressure and myogenic tone.

For the shake of the authors and the readers I include here a list of questions that may further support the development of this story significantly increasing its coherence and cohesion:

1) This Reviewer is concerned with the usage of the transgenic mouse model as a genetic mouse model lacking RNA editing in Filamin A. Given that the deleted intronic region may be important for the RNA processing of Filamin A or for the secondary structure and normal protein function outside the RNA editing event and subsequent amino acid exchange, the authors may consider validating their results by inserting a point mutation at Q2341R (as also implied by the Reviewer 3) using the new CRISPR/Cas type IV technology. In order to avoid the time consuming nature of creating the more appropriate mouse strain, the authors may just validate their most important findings in vascular smooth muscle cells *in vitro*. Further, representative western blots of the Filamin A expression in primary vascular smooth muscle cells should be depicted in Figure 2D. RNA-sequencing experiments of isolated mouse vascular smooth muscle cells showing the alignment of

the reads of Filamin A to its expected genomic sequence would help to exclude any potential RNA processing effect on Filamin A pre-mRNA.

2) Mass spectrometry experiments showed that the mutant Filamin A (unedited) differentially binds to 50 proteins, most of them being nuclear factors. Firstly, this result needs to be validated in at least three biological experiments (and not just technical replicates) after immunoprecipitation of the pre-edited Q2341R mutant Filamin A vs. the unedited form. Further, the authors shall apply a computational strategy to delineate the exact mechanism linking the effect of RNA editing (re-coding of the protein) with the cellular phenotypes (phosphorylation of myosin light chain, effect to the levels of RhoA-GTP, phosphorylation of myosin light chain phosphatase 1 (MYPT1) and of CPI-17). The study would be significantly strengthened if the underlying mechanism is revealed.

3) Is ADAR2 binding to Filamin A pre-mRNA abolished in mvSMCs after deletion of the Filamin A ECS? Where exactly does ADAR2 bind to Filamin A? How the deletion of the ECS affect ADAR2 binding and editing of exon 42 adenosine? iCLIP experiments may help exactly map the binding of ADAR2 to Filamin A. In general the role of ADAR2 in the described phenotypes has not been studied. The authors may consider evaluating the role of ADAR2 in the mvSMC phenotypes.

4) How do the authors explain their findings considering the lack of any effect on blood pressure in the ADAR2 knockout mice as previously reported (Horsch M et al., J Biol Chem. 2011 May 27;286(21):18614-22)?

5) The authors report a decrease of RNA editing rate in individuals with a positive history of cardiovascular disease. This association may be interesting but does not prove any causative relationship between cardiovascular disease and RNA editing of Filamin A. How are the ADAR2 expression levels in these tissue probes? A histological analysis of the tissue specimens shown in Figure 1E for vascular disease and SMC-related pathologies is integral for the correct interpretation of the findings. Subjects with a positive history of cardiovascular disease do not necessarily have a diseased tibial artery or aorta. If the authors wish to make any claim regarding the clinical relevance of the RNA editing of Filamin A, then this should be studied in real diseased tissue probes (for instance: aortic aneurysm tissues, atherosclerotic plaques, lung arterioles from patients pulmonary arterial hypertension, left ventricle biopsies from patients with ischemic or dilative cardiomyopathy, etc). The characteristics of the patients shall be also shown and a multivariable analysis shall be applied showing the independent association of Filamin A RNA editing rate with (cardio-)vascular disease. The current reported findings, although interesting, are not conclusive.

6) Very often the interpretation of the reported findings exceeds the scope or the real result of the experiment. For instance the title of the manuscript says that Filamin A RNA editing regulates diastolic blood pressure and cardiovascular remodeling, while:

- a) all mouse experiments were done in 4-6-month old male mice neglecting the absence of any effect in earlier age or in female mice (which were not studied at all).
- b) the effect in the diastolic pressure is only marginal, possibly only reaching statistical significance due to the outlier in the WT group
- c) there is no effect in systolic blood pressure
- d) there is hardly any clinical relevance of an isolated diastolic pressure mechanism, which is unlikely to cause heart failure alone as implied in this manuscript
- e) the renin-angiotensin-aldosterone system is not studied (main hormones regulating blood pressure)
- f) the marginal effect in the diastolic blood pressure cannot fully explain the effect size in the left ventricular mass as depicted in Figure 6E
- g) the adventitial thickening of the aortic wall is not related to the other mouse phenotypes and especially with the diastolic pressure
- h) the perivascular fibrosis is not explained by the depicted SMC phenotypes
- i) the left (? what about the right?) ventricular wall thickness is not confirmed by cardiac MRI
- j) the phenotypes presented in Figure 6 are not proven to derive from the SMC contraction phenotype

For all these reasons, the authors may consider adapting accordingly their title of the manuscript and their "generous" interpretations throughout the manuscript.

7) If the authors believe that the effect of the unedited Filamin A in diastolic pressure is

reproducible, then instead of the aorta adventitia they shall examine the resistance arterioles which are the main regulators of diastolic pressure. Does Filamin A affect the wall thickness and the diameter of the lumen of the resistance arterioles?

8) How do the authors explain the severe cardiovascular phenotypes (aortic aneurysm, heart failure) in the 24-month old mice? What are the underlying mechanisms? How is the blood pressure at this time point? Why are these phenotypes only observed in some animals and not all animals? This Reviewer believes that Filamin A plays probably a very important role beyond the SMCs in the development of heart failure and thus it shall be studied carefully after dissection of the cellular origin.

Minor comments:

1) Is there any correlation between ADAR2 and Filamin A gene expression as implied in Figure 1A and B?

2) The authors claim that Filamin A is the main ADAR2 substrate. However they do not show any experiments supporting this notion apart from the isolated RNA editing rate. Further the glutamate receptor GluR-B is known to be edited up to 100% (Higuchi M, et al., Nature. 2000 Jul 6;406(6791):78-81) while absence of RNA editing of this receptor results in postnatal lethality. Thus, the author shall reconsider adapting their conclusions regarding which substrate of ADAR2 is the most important for life.

3) Figure 1D: upper graph: the Y axis shall start from "0" instead of "40".

4) Fig 4B: total RhoA levels seem to differ. Please provide the tubulin levels as well.

5) Is Filamin A ubiquitously expressed? How do the authors exclude any other effect from cells other than SMCs in their described mouse phenotypes?

6) page 7, last paragraph: the SD shall be added to the percentages of RNA editing rate

7) Why did the authors treat the aortic rings with the thromboxane A2 receptor agonist U46619? Does the unedited Filamin A affect the vSMC pressure-overload response?

7) Does Filamin A affect SMC relaxation as implied in the manuscript or only the contraction?

8) Verification of SMC isolation procedure shall be shown (FACS for SMA, cell culture photos, etc). How a contamination with fibroblasts was excluded in the cell culture experiments?

9) Is calcium signaling affected by Filamin A?

10) page 16, first paragraph: "Also, several genes change in their expression in the FLNA mice". Which are these genes? Are they involved in the described phenotypes?

11) The heart function (LVEF, LVEDV, LVESV, LV mass) in young and old mice shall be shown in the main figures of the manuscript. Figure S11 is not convincing mainly due to the very low number of mice included in each observation.

2nd Revision - authors' response

18th May 2018

Response to reviewers

Reviewer 1:

This manuscript demonstrates a novel contribution of RNA-editing, specifically in filamin A pre-mRNA in controlling blood pressure. The authors have shown that lack of filamin A pre-mRNA editing leads

to increased phosphorylation of myosin light chain and an increase in RhoA/Rock and PLC/PKC signaling, which could explain the increased smooth muscle contraction in FLNA Δ ECS vSMCs. The experiments show convincing data about the in vivo vasculopathy in FLNA Δ ECS mice and mass spec analysis shows that unedited FLNA binds stronger to proteins involved in the contractile machinery, which could be one of the mechanisms. It is interesting that the human GTEx data set showing decreased RNA editing was analyzed from dilated cardiomyopathy patients where the pathology is a primary cardiomyopathy (cardiomyocyte disease), but the FLNA Δ ECS mice show only secondary cardiomyocyte hypertrophy and the phenotype is rather in smooth muscle cells, demonstrating diastolic hypertension. Indeed smooth muscle cells in the heart could be contributing to the phenotype in humans or it could be another mechanism or just the difference between species. I think the manuscript will be interesting enough to investigators studying smooth muscle cell biology and vasculopathy. I just have a minor concern about the figure.

Minor concern

1. Fig7A is a little confusing to me. The number of WT mice are explained as 4 in the manuscript but the figure seems to have 5 solid lines representing WT. Also I do not see a pink solid line for an outlier, it rather looks purple to me if I am understanding the figure correctly.

We are sorry for this confusion. Indeed, the pink/purple solid line belongs to the 5th FLNA Δ ECS mouse, it is rightly pointed out to be confusing and we have changed this line to a dotted purple line.

Reviewer 2:

The submitted revised manuscript entitled "RNA editing of Filamin A pre-mRNA regulates diastolic blood pressure and cardiovascular remodeling" by Jain M, et al. describes the extent of RNA editing of Filamin A in various human tissues ranging between 8 and 92% based on an RNA editing analysis of the Genotype-Tissue Expression project (GTEx) as well as in human tissue specimens from the tibial artery and the aorta of control subjects and patients with a positive history of a cardiovascular disease. The authors report a decrease of about 15-20% of the extent of RNA editing rate of Filamin A in aortic-arterial tissues from subjects with a positive history of cardiovascular disease.

Mechanistically the authors created a transgenic mouse strain by deleting a 228 bp long intronic region located in intron 42-43 which works as the editing complementary site (ECS) of the double-stranded RNA formed by this region and the exon 42. It is known that RNA editing of filamin A pre-mRNA at a specific adenosine in the exon 42 leads to a Q/R substitution in Ig-repeat 22 of the encoded protein. The authors show that the deletion of the 228bp intronic ECS results in absence of RNA editing of the specific exon 42 adenosine. The aortic ring contraction from mice deficient for the Filamin A ECS was found to be up to 25% increased after treatment with the thromboxane A₂ receptor agonist U46619 compared to WT aortic rings. Mouse studies utilizing only 16-24 week old male mice revealed a slight increase of around 5-8 mmHg the diastolic arterial pressure in the resting, but not in active phase. Histological analysis of the aorta cross-sections revealed a thicker adventitial area in the Filamin A transgenic mice. Heart sections showed an increase in perivascular fibrosis and a significant increase in the left ventricular wall thickness as well as of the cardiomyocyte cross-sectional size. Mechanistically, the authors show that deletion of the Filamin A ECS in murine primary aortic smooth muscle cells induces the phosphorylation of myosin light chain, the levels of RhoA-GTP, the phosphorylation of myosin light chain phosphatase 1 (MYPT1) and of CPI-17, which are all critical factors for smooth muscle cell contraction. Inhibition of either the ROCK or the PKC signaling abolished the CPI-17 effect. Further, the authors show that the localization of p190-RhoGAP is affected. Interestingly, the authors performed a Filamin A immunoprecipitation followed by mass spectrometry in three technical replicates showing that from the 300 proteins, 20 were enriched in the mutant (unedited) Filamin A, while 30 were enriched in the WT (edited) Filamin A form. Last, the authors performed cardiac MRI showing that the velocity in the ascending aorta was decreased in the Filamin A transgenic mice during the systolic phase.

Although the authors should be admittedly congratulated for the extent of their efforts to analyse in depth the effect of RNA editing of Filamin A in the cardiovascular homeostasis, the study unfortunately lacks scientific coherence and cohesion and the presented very interesting findings are poorly connected with each other. Further, the deletion of a 228 bp intronic area instead of the specific replacement of only the nucleotide of interest responsible for the amino acid exchange (Q2341R) limits the interpretation and relevance of the results. The revised manuscript does not fully address the concerns of the Reviewer 1 related to the end-organ effects of arterial hypertension and of the Reviewer 3 related to the absence of a mechanism related to RNA editing that connects the described phenotypes. The dynamic landscape of RNA editing of repetitive elements and coding regions including Filamin A in human tissues of the GTEx has been recently reported (Tan MH et al.,

Nature. 2017 Oct 11;550(7675):249-254). The role of Filamin A in smooth muscle cell function has been also recently reported (Retailleau K et al., Cell Rep. 2016 Mar 8;14(9):2050-2058) showing a distinct differential role of Filamin A in basal blood pressure and myogenic tone.

We appreciate that this reviewer recognizes the extent of analysis that was performed in this study. We agree that not all points in the complex puzzle of how editing affects smooth muscle contraction are fully understood. Admittedly, our mass spectrometry data identifies several candidates that may be linked to smooth muscle cell contraction. Testing all of them by mapping and disrupting their interaction with FLNA would take years. Still, we believe that the findings presented by us are of interest to a wide readership and will stimulate further research in this unexplored area.

The reviewer is correct that the editing landscape of coding substrates in the human has recently been reported. However, from the mentioned study from the Li lab, it is not apparent that editing is a) most abundant in the cardiovascular system, b) that FLNA is the single most edited transcript in the human transcriptome and c) that in a cohort of cardiovascular disease patients editing drops significantly.

The reviewer is also correct that the involvement of Filamin A in smooth muscle contraction has been identified. This was shown by a complete deletion of Filamin A. Obviously, as an actin crosslinking protein, loss of filamin impacts mechanosensing of cells. However, loss of FLNA is not a physiological condition. Our study shows, on the other hand, that the function of FLNA can be regulated by a single RNA editing event and that editing levels can vary.

For the sake of the authors and the readers I include here a list of questions that may further support the development of this story significantly increasing its coherence and cohesion:

1) This Reviewer is concerned with the usage of the transgenic mouse model as a genetic mouse model lacking RNA editing in Filamin A. Given that the deleted intronic region may be important for the RNA processing of Filamin A or for the secondary structure and normal protein function outside the RNA editing event and subsequent amino acid exchange, the authors may consider validating their results by inserting a point mutation at Q2341R (as also implied by the Reviewer 3) using the new CRISPR/Cas type IV technology. In order to avoid the time consuming nature of creating the more appropriate mouse strain, the authors may just validate their most important findings in vascular smooth muscle cells in vitro. Further, representative western blots of the Filamin A expression in primary vascular smooth muscle cells should be depicted in Figure 2D. RNA-sequencing experiments of isolated mouse vascular smooth muscle cells showing the alignment of the reads of Filamin A to its expected genomic sequence would help to exclude any potential RNA processing effect on Filamin A pre-mRNA.

The codons encoding glutamine are CAG and CAA, in order to make them uneditable, one would have to remove the central A from them, which is obviously not possible without recoding therefore, crispr/cas cannot be used to eliminate editing of FLNA.

This reviewer is obviously concerned that traditional homologous recombination technology is causing more side effects than CRISPR-based manipulations. Obviously, we have backcrossed our mice >5x to minimize the problem of off-target effects after homologous recombination in ES cells.

The reviewer seems also concerned that FLNA expression levels have changed upon manipulation of the intronic sequence. To assure that this is not the case we have performed a) qPCR (Figure 2c) and Western Blot comparison (Figure 2d). We also have performed mass spec analysis (Figure EV4)- if FLNA expression was changed, we would observe a general trend of all proteins changing their relative abundance relative to FLNA. This is not the case, instead, we see some proteins showing an increased, others showing a decreased co-purification with FLNA, depending on its editing status. Moreover, we have also performed RNA seq on tissues. The relative position of the FLNA mRNA amongst the bulk of all other RNAs has never changed (not shown).

Still we have now added a western blot showing FLNA expression in stomach tissue to Figure 2 D and FLNA expression in vSMCs in Figure S5b.

For other purposes we have also done RNA seq of 3 replicates of the colon, where FLNA editing is also high. We have not observed any significant change in the expression of FLNA between wt and FLNA-ECS cells. We show this now in supplementary figure S4a.

2) Mass spectrometry experiments showed that the mutant Filamin A (unedited) differentially binds to

50 proteins, most of them being nuclear factors. Firstly, this result needs to be validated in at least three biological experiments (and not just technical replicates) after immunoprecipitation of the pre-edited Q2341R mutant Filamin A vs. the unedited form. Further, the authors shall apply a computational strategy to delineate the exact mechanism linking the effect of RNA editing (re-coding of the protein) with the cellular phenotypes (phosphorylation of myosin light chain, effect to the levels of RhoA-GTP, phosphorylation of myosin light chain phosphatase 1 (MYPT1) and of CPI-17). The study would be significantly strengthened if the underlying mechanism is revealed.

Maybe we were unprecise by calling our mass spec experiment a technical replicate: We have performed three independent immunoprecipitation experiments for wt and FLNA Δ ECS cells grown on different dates. We call this technical replicate because the cells were derived from the same mouse (and obviously we only have one isogenic transgenic mouse line). Strictly speaking, one would maybe have to make three transgenic mouse lines and perform the experiments independently. We also repeated the experiments in cell lines expressing only edited or only unedited FLNA (in M2 cells which lack endogenous FLNA). By en large we also found the same shifts in interactions. However, as the levels of FLNA expression is variable in those cell lines results are harder to normalize and to interpret which is why we preferred not to include the data in our present manuscript.

Unfortunately we do not understand the request for a computational strategy that will delineate the exact molecular mechanisms.

3) Is ADAR2 binding to Filamin A pre-mRNA abolished in mvSMCs after deletion of the Filamin A ECS? Where exactly does ADAR2 bind to Filamin A? How the deletion of the ECS affect ADAR2 binding and editing of exon 42 adenosine? iCLIP experiments may help exactly map the binding of ADAR2 to Filamin A. In general the role of ADAR2 in the described phenotypes has not been studied. The authors may consider evaluating the role of ADAR2 in the mvSMC phenotypes.

The double stranded structure formed by the intronic editing complementary sequence (ECS) and the exonic editing site has been described in our 2004 paper (Levanon, Hallegger, et al NAR 2004). Obviously, ADAR2 will bind to this double stranded structure. Deletion of the ECS makes the structure single stranded (we can show folding predictions for this). It is clear that altering the double stranded structure formed between intron and exon 42 will disrupt ADAR2 binding. However, as we can show that neither RNA nor protein levels of FLNA change (see point 2 above) the binding or lack of binding of ADAR2 onto the FLNA pre-mRNA has apparently no impact on mature protein levels.

4) How do the authors explain their findings considering the lack of any effect on blood pressure in the ADAR2 knockout mice as previously reported (Horsch M et al., J Biol Chem. 2011 May 27;286(21):18614-22)?

It is indeed interesting that our phenotype was not noted in the Horsch et al paper of 2011 where a full phenotyping of an ADAR2 deficient mouse-rescued with GluRB, pre-edited- was made. Mrs. Horsch works at the German Mouse Clinic and also our mice were initially phenotyped at the german mouse clinic. At the mouse clinic the blood pressure phenotype of our mice went unnoticed. However, the perivascular fibrosis phenotype – Figure 6c, was noticed at the German Mouse Clinic. This also explains the extended list of authors on our manuscript. Why did the blood pressure phenotype go unnoticed at the GMC? The GMC performs tail cuff blood pressure measurements. Obviously, the mice are wide alert during this measurement. Blood pressure changes during resting phases can only be detected by permanently implanted devices. We believe that the differences in measuring technologies explains why the German mouse clinic failed to detect an elevated blood pressure both in the ADAR2 and the FLNA Δ ECS mice. The analysis of the ADAR2 deficient mouse by Horsch et al noticed a lower heart rate and shorter inspiratory rate in editing deficient mice (only in male cohorts) which could in fact indicate a cardiac problem. To comment on the general function of ADAR2 on vSMC function: ADAR2 also edits Filamin B that can also heterodimerize with filamin A, it also edits several other targets that may affect cell contraction. We agree that understanding the global impact of ADAR2 on cardiovascular integrity would be valuable, similar to the study by Stellos et al on ADAR1. However, in the present paper we wanted to decipher the impact of a single RNA-editing event and not extend the study to at least tens of substrates.

5) The authors report a decrease of RNA editing rate in individuals with a positive history of cardiovascular disease. This association may be interesting but does not prove any causative relationship between cardiovascular disease and RNA editing of Filamin A.

We did not mean to make a causal connection between these two events co-occurring in humans. This is why we made the mouse, to test whether an observed correlation may also have an underlying cause. The corresponding section in the manuscript reads; We then sought to determine whether changes in filamin A editing have a causal effect on the development of cardiovascular pathologies. We generated a mouse with an exclusive deficit in FLNA editing.

How are the ADAR2 expression levels in these tissue probes?

We have performed qPCR to determine the ADAR2 expression levels in these tissues. No correlation between ADAR2 expression and FLNA editing levels were observed in our samples. This is now shown in Supplementary Figure S3C.

A histological analysis of the tissue specimens shown in Figure 1E for vascular disease and SMC-related pathologies is integral for the correct interpretation of the findings. Subjects with a positive history of cardiovascular disease do not necessarily have a diseased tibial artery or aorta. If the authors wish to make any claim regarding the clinical relevance of the RNA editing of Filamin A, then this should be studied in real diseased tissue probes (for instance: aortic aneurysm tissues, atherosclerotic plaques, lung arterioles from patients pulmonary arterial hypertension, left ventricle biopsies from patients with ischemic or dilative cardiomyopathy, etc).

We are sorry for not being precise in describing the samples for 1E. These were taken from fresh corpses donated to anatomy courses. The hearts of these donors were visually inspected for signs of hypertrophy by a clinically trained anatomist. Donors with hearts showing clear signs of hypertrophy and a septum thickness of more than 11mm were considered as diseased, while corpses where the hearts showed no signs of hypertrophy served as controls. Obviously, we cannot exclude that "control" patients also had early signs of cardiovascular disease that did not yet manifest themselves in hypertrophy. Dorsal aortae and tibial arteries of these donors were isolated and a small piece was used for RNA isolation and editing analysis. We were very surprised to see a strong correlation between visual hypertrophy of the heart and decreased editing levels as seen in figure 1E. This reviewer is correct by stating that cardiovascular disease may not necessarily correlate with diseased arteries and aortae.

*In fact, figure 1E also shows that the editing levels of at least 20% of all samples originating from diseased donors lie within the normal range. This indicates that decreased editing **may** lead to cardiovascular disease but also that **not every cardiovascular disease is associated with a drop in editing**, just as suspected by this reviewer. We have now clearly stated this in the section describing Figure 1E.*

The patients characteristics of the patients shall be also shown and a multivariable analysis shall be applied showing the independent association of Filamin A RNA editing rate with (cardio-)vascular disease. The current reported findings, although interesting, are not conclusive.

We are sorry but a complete health record is neither available for the people in the Schafer et al study, nor from the body donations investigated. We do not doubt that also other factors may affect cardiovascular health. The focus of this study was on the function of FLNA editing, not on the identification of all cardiovascular risk factors.

- 6) Very often the interpretation of the reported findings exceeds the scope or the real result of the experiment. For instance the title of the manuscript says that Filamin A RNA editing regulates diastolic blood pressure and cardiovascular remodeling, while:
- a) all mouse experiments were done in 4-6-month old male mice neglecting the absence of any effect in earlier age or in female mice (which were not studied at all).

The reviewer may have overlooked our study of younger mice: On page 13-14 we state "To test, whether the observed cardiac phenotypes were a secondary effect to hypertension or would also appear autonomously, we tested younger 21 day old mice for the appearance of either perivascular fibrosis or enlarged cardiomyocytes. However, in contrast to 5 months old mice, no signs of abnormal cardiac organization were detected (Fig. S6)" and accordingly show that the cardiomyocyte diameter is secondary and not primary in Fig S6. Apparently this paragraph escaped the attention of this reviewer.

We also studied the blood pressure of female mice, which was also elevated. Also this is included in figure 5. As can be seen, the female samples fall right in between the male samples. This was possibly overlooked by this reviewer.

b) the effect in the diastolic pressure is only marginal, possibly only reaching statistical significance due to the outlier in the WT group

*We understand this reviewer's concern. However, the significance observed is not the result of the outlier, as a larger SD actually weakens rather than strengthens the pvalue. In a previous version of this manuscript we had only analyzed four wild-type and four mutant **male** mice. There, the difference between mutant and wt mice was around 9 mm Hg and had a significance of $p=0.012$. Now, a total of 18 mice were analyzed 5 males and **4 female** for both wild-type and mutant mice. This now included the "outlier" mouse. Interestingly, the p-value stayed the same. I.e. the larger sample size did not lead to a better p value because the outlier was included.*

c) there is no effect in systolic blood pressure

Yes, at least from the number of mice and the technology used by us, we could not detect significant changes in the systolic blood pressure.

d) there is hardly any clinical relevance of an isolated diastolic pressure mechanism, which is unlikely to cause heart failure alone as implied in this manuscript

This is a valuable argument. Indeed, our mice do not consistently display heart failure and we did not claim this in the manuscript. Instead, we state that our mice start developing left ventricular hypertrophy but do not die prematurely. We also agree that the change in diastolic blood pressure will not be a sole factor leading to cardiac failure. We simply state that blood pressure and cardiovascular remodeling is affected, which we can document by our experiments.

e) the renin-angiotensin-aldosterone system is not studied (main hormones regulating blood pressure)

Yes, we did not study the hormonal system controlling blood pressure. The focus of this study was on the effect of RNA editing on cardiovascular function, a previously unnoticed phenomenon. We feel that including all possible pathways would go beyond the scope of this manuscript.

f) the marginal effect in the diastolic blood pressure cannot fully explain the effect size in the left ventricular mass as depicted in Figure 6E

*We agree that most likely other factors are contributing to the observed phenotypes. We have now stated in the discussion that our data show that FLNA editing **can contribute** to cardiovascular health.*

g) the adventitial thickening of the aortic wall is not related to the other mouse phenotypes and especially with the diastolic pressure

This is correct. Still, as this is a phenotype that affects the cardiovascular system we would prefer to keep this figure in the manuscript.

h) the perivascular fibrosis is not explained by the depicted SMC phenotypes

Increased perivascular fibrosis has been correlated with ROCK activity and a decrease in ROCK activity was shown to reduce perivascular fibrosis Circulation. 2005 Nov 8; 112(19): 2959–2965. As we observe an increase in ROCK activity, this can be linked to the observed perivascular fibrosis Circ J. 2016 Jun 24;80(7):1491-8. doi: 10.1253/circj.CJ-16-0433. We have included these references and mention their content in the results to figure 6.

i) the left (? what about the right?) ventricular wall thickness is not confirmed by cardiac MRI

We only observe left ventricular thickening but no right ventricular abnormalities by histology. For financial reasons we have not done the same number of mice by MRI as we did by histology. By MRI we only observe an increase in the left ventricular wall for a sample size of 4 wt and 4 mutant mice reaching a ~8% increase in LV wall area with a significance of only $p=0.3$. Growing mice to the same cohort size and age and to repeat the experiment would go beyond our current capacity.

j) the phenotypes presented in Figure 6 are not proven to derive from the SMC contraction

phenotype

For all these reasons, the authors may consider adapting accordingly their title of the manuscript and their "generous" interpretations throughout the manuscript.

We have considered this reviewer's comment and have modified the title "RNA editing of filamin A pre-mRNA regulates vascular contraction and diastolic blood pressure and cardiovascular remodeling". We have also tried to eliminate all parts in the manuscript that could be seen as an overinterpretation of our manuscript.

7) If the authors believe that the effect of the unedited Filamin A in diastolic pressure is reproducible, then instead of the aorta adventitia they shall examine the resistance arterioles which are the main regulators of diastolic pressure. Does Filamin A affect the wall thickness and the diameter of the lumen of the resistance arterioles?

This is a good suggestion. We have stained the resistance arterioles of the kidney (See supplementary figure S7). However, we did not observe a wall thickening in these vessels. Puzzled by this observation we went back to our old study (Stulic and Jantsch, 2013) and found that editing in the kidney -not knowing which tissue is affected there- reaches only 20%. It might be, that small arterioles show little or no editing, therefore not having an impact on vessel contraction.

8) How do the authors explain the severe cardiovascular phenotypes (aortic aneurysm, heart failure) in the 24-month old mice? What are the underlying mechanisms? How is the blood pressure at this time point? Why are these phenotypes only observed in some animals and not all animals? This Reviewer believes that Filamin A plays probably a very important role beyond the SMCs in the development of heart failure and thus it shall be studied carefully after dissection of the cellular origin.

We agree that the data on the 24 months old mice is not easily explicable, especially since penetrance is not at 100%. Consequently, the data is also not statistically sound. We also have no blood pressure data on these mice. The blood pressure recorders are not kept subcutaneously for more than a few weeks. Mice start to scratch and injure themselves after carrying the blood pressure monitors which is why they are sacrificed. The aged mice had no blood pressure implants. We included these data in the original manuscript as we thought it shows an interesting aspect. However, as this is not investigated thoroughly we have removed this part from the manuscript.

Minor comments:

1) Is there any correlation between ADAR2 and Filamin A gene expression as implied in Figure 1A and B?

Figures 1A and 1B show that ADAR2 expression and Filamin A expression are highest in the vasculature and in the smooth muscle layer of some gastrointestinal tissues. We now state this in the manuscript.

2) The authors claim that Filamin A is the main ADAR2 substrate. However they do not show any experiments supporting this notion apart from the isolated RNA editing rate. Further the glutamate receptor GluR-B is known to be edited up to 100% (Higuchi M, et al., Nature. 2000 Jul 6;406(6791):78-81) while absence of RNA editing of this receptor results in postnatal lethality. Thus, the author shall reconsider adapting their conclusions regarding which substrate of ADAR2 is the most important for life.

The reviewer must have seen a previous version of our manuscript for another journal and possibly copied this comment? Nowhere do we state that FLNA is the substrate "most important for life". However, we do state that FLNA is the most abundant substrate, which is clearly documented by our data in Figure 1. If the most abundant substrate is edited to almost 90% this makes it also the main substrate of the enzyme.

3) Figure 1D: upper graph: the Y axis shall start from "0" instead of "40".

We have changed this.

4) Fig 4B: total RhoA levels seem to differ. Please provide the tubulin levels as well.

We have added this

5) Is Filamin A ubiquitously expressed? How do the authors exclude any other effect from cells other than SMCs in their described mouse phenotypes?

Yes, Filamin is ubiquitously expressed but as Figure 1A shows, it is most highly expressed in the vasculature. Also editing of Filamin A is highest in the vasculature. Within the vasculature, editing is highest in SMCs. However, we nowhere state that the cardiovascular remodeling is exclusively due to editing in SMCs. We only show that vSMCs show the same hypercontraction phenotype as aortae.

6) page 7, last paragraph: the SD shall be added to the percentages of RNA editing rate

We have done this.

7) Why did the authors treat the aortic rings with the thromboxane A2 receptor agonist U46619? Does the unedited Filamin A affect the vSMC pressure-overload response?

We have used the U46619 as this showed the most reproducible effect in our contraction assays. We cannot comment on the pressure-overload response.

7) Does Filamin A affect SMC relaxation as implied in the manuscript or only the contraction?

Filamin A editing does affect relaxation on precontracted aortae but we have not extensively studied this aspect. The corresponding SMC relaxation is shown in figure EV2b.

8) Verification of SMC isolation procedure shall be shown (FACS for SMA, cell culture photos, etc). How a contamination with fibroblasts was excluded in the cell culture experiments?

We mention this in the Methods section: "The cells were then immortalized and purified from fibroblasts using a negative Magnetic activated cell sorting (MACS) selection with CD90.1 microbeads" We have also provided a Supplementary figure S5 showing the enrichment.

9) Is calcium signaling affected by Filamin A?

We show that BAPTA abrogates the hypercontraction phenotype in figure EV2E and also state that Ca⁺⁺ signaling must be involved (page 10)

10) page 16, first paragraph: "Also, several genes change in their expression in the FLNA mice". Which are these genes? Are they involved in the described phenotypes?

To our knowledge these genes are not involved in the observed phenotype. We have collected NGS data from the colon, where FLNA editing is similarly high. We are still analyzing these data for another colon specific phenotype we are studying.

11) The heart function (LVEF, LVEDV, LVESV, LV mass) in young and old mice shall be shown in the main figures of the manuscript. Figure S11 is not convincing mainly due to the very low number of mice included in each observation.

We agree that previous figure S11 was not convincing as we do not have a high number of mice at that age. We have therefore removed this part from the manuscript.

3rd Editorial Decision

22nd June 2018

Thank you for submitting your revised manuscript to the EMBO Journal. I have now gone through the new version and your response to the referee concerns from the previous round and I am glad to inform you that your manuscript is now in principle accepted for publication here.

However, there are still a couple of formatting issues that need to be sorted out before I can officially accept the manuscript and transfer it to production. I therefore have to ask you to submit one final revision that addresses the following points:

Accepted

27th June 2018

Thanks for submitting the final version of your manuscript, I am pleased to inform you that your study is now officially accepted for publication in The EMBO Journal.

Corresponding Author Name: Michael F. Jantsch

Manuscript Number: EMBOJ-2016-94813R